# On the nanoscale structural evolution of solid discharge products in lithium-sulfur batteries using operando scattering

Christian Prehal [1] ✉, Jean-Marc von Mentlen [1], Sara Drvarič Talian [2], Alen Vizintin [2], Robert Dominko [2,3], Heinz Amenitsch[4], Lionel Porcar[5], Stefan A. Freunberger [6] ✉ & Vanessa Wood [1] ✉

The inadequate understanding of the mechanisms that reversibly convert molecular sulfur (S) into lithium sulfide ($Li_2S$) via soluble polysulfides (PSs) formation impedes the development of high-performance lithium-sulfur (Li-S) batteries with non-aqueous electrolyte solutions. Here, we use operando small and wide angle X-ray scattering and operando small angle neutron scattering (SANS) measurements to track the nucleation, growth and dissolution of solid deposits from atomic to sub-micron scales during real-time Li-S cell operation. In particular, stochastic modelling based on the SANS data allows quantifying the nanoscale phase evolution during battery cycling. We show that next to nano-crystalline $Li_2S$ the deposit comprises solid short-chain PSs particles. The analysis of the experimental data suggests that initially, $Li_2S_2$ precipitates from the solution and then is partially converted via solid-state electroreduction to $Li_2S$. We further demonstrate that mass transport, rather than electron transport through a thin passivating film, limits the discharge capacity and rate performance in Li-S cells.

Lithium-sulfur (Li-S) batteries are considered strategic candidates to reduce the environmental impact of non-aqueous Li-based batteries[1]. The high expectations arise from the large theoretical capacities, abundance, and low cost of sulfur[2–4]. Li-S batteries reversibly cycle sulfur to lithium sulfide (S / $Li_2S$), typically in a highly porous carbon cathode soaked with a liquid, non-aqueous electrolyte and using a lithium metal anode. Discharge converts S to $Li_2S$ stepwise via polysulfides (PSs) $Li_2S_x$ (2 < x < 8). Practical realization of Li-S cells is hindered by incomplete S utilization, poor S/$Li_2S$ mass loadings, rapid capacity fading, low rate capabilities, and irreversible reactions of PSs at the anode[3,5,6]. These issues all trace back to insufficient understanding of S-to-$Li_2S$ conversion.

The physical-chemical mechanism to reversibly form and dissolve solid $Li_2S$ remains controversial[7,8]. Many studies consider $Li_2S$ to form via direct electroreduction of $Li_2S_2$ or longer-chain PSs at the carbon-electrolyte interface[8–12]. However, as the electrodeposition of an electronic insulator like $Li_2S$ is in principle self-limited, the fact that $Li_2S$ deposits are beyond tens and hundreds of nm in size and porous[13–15] suggest that they form via a solution-mediated process. This is supported by the finding that capacity is limited by mass transport rather than electron transport through a passivating film[16–18]. Such a solution-mediated processes could be the disproportionation of dissolved PSs, as considered by some studies[13,16,19]. Another option would be direct electroreduction of molecular $Li_2S_2$ to dissolved $Li_2S$ (2 $Li^+$, $S^{2-}$), which

[1]Department of Information Technology and Electrical Engineering, ETH Zürich, Gloriastrasse 35, 8092 Zürich, Switzerland. [2]Department of Materials Chemistry, National Institute of Chemistry, Hajdrihova 19, 1000 Ljubljana, Slovenia. [3]Faculty of Chemistry and Chemical Technology University of Ljubljana, Večna pot 113, 1000 Ljubljana, Slovenia. [4]Institute for Inorganic Chemistry, Graz University of Technology, Stremayrgasse 9, 8010 Graz, Austria. [5]Institut Laue–Langevin, 71 Avenue des Martyrs, Grenoble 38042, France. [6]Institute of Science and Technology Austria (ISTA), Am Campus 1, 3400 Klosterneuburg, Austria. ✉e-mail: cprehal@ethz.ch; stefan.freunberger@ist.ac.at; vwood@ethz.ch

then precipitates solid $Li_2S$ crystallites, similar to the electrodeposition of $NaO_2$ or $KO_2$ in $Na-O_2$ and $K-O_2$ batteries[20]. However, large deposits beyond tens or hundreds of nanometers would require a solubility of $Li_2S$ beyond the reported $10^{-6}$ M[15,21]. Some studies consider solid $Li_2S_2$ to be involved, however, there is little experimental evidence[22–24].

While operando X-ray diffraction[25–27] and spectroscopy[28–31] provide insights into the chemistries occurring during (de)lithiation, a complete understanding of the mechanisms of $Li_2S$ formation requires a detailed chemical as well as structural picture on atomic and nanometer length scales. The structures within Li-S cells have been studied using (operando) electron and X-ray microscopy[32–35]. (Operando) microscopy provide unique model-free structural information, but these techniques can be limited by, the $Li_2S$ stability, the resolution, field of view, the challenges of 3D imaging or the cell design. The cell design in an operando nanotomography cell, for example, requires extremely small electrodes and separators in the order of a few tens of micrometers[36]. Operando transmission electron microscopy (TEM) cells are typically realized with a solid ($Li_2O$) electrolyte[37]. The mechanism of $S/Li_2S$ conversion in such all-solid-state battery is different from Li-S batteries with non-aqueous liquid electrolyte solutions. Small angle scattering can provide complementary structural sensitivity from sub-nm to 100 nm, regardless of whether the probed phases are crystalline, amorphous or liquid[38]. As an additional advantage, the used operando cells have often a design and electrochemical characteristics similar to conventional lab-scale coin-cells[39–42]. Recent operando small angle neutron and X-ray scattering studies confirmed the ability to follow the morphological evolution of solid discharge products not much larger than a few nm[41,42]. Neutron and X-ray scattering are complementary methods since the phases are probed with different scattering contrasts.

Here, we perform operando small and wide angle X-ray scattering (SAXS/WAXS) and operando small angle neutron scattering (SANS) to gain simultaneous structural and chemical insights from atomic to sub-micrometer scales with time resolutions down to several seconds[39,40,43]. Stochastic modelling enables quantitative interpretation of the SANS results[28,41]. During lithiation, we observe the formation of a hierarchical structure, consisting of aggregates of $Li_2S$ crystallites and a second solid short-chain PS phase, which we argue to be $Li_2S_2$. $Li_2S$ is formed by the solid-state conversion of $Li_2S_2$. During delithiation, the reverse process occurs. Complementary information from Raman spectroscopy, electron microscopy, and electrochemical measurements allows us to validate our model for (de)lithiation. These findings show that discharge capacities and rates in Li-S batteries are limited by transport through the tortuous solid deposits and give inspiration for how cell design, electrolyte selection, and cycling proposals can be used to optimize performance[17].

## Results

### Operando small and wide angle X-ray scattering measurements

Operando SAXS/WAXS experiments are carried out with a commercial (SAXS/WAXS) electrochemical operando scattering cell[40] holding a high surface area conductive carbon cathode (carbon black with specific surface area = $1400 \, m^2 \, g_C^{-1}$), a Li metal anode, and a catholyte comprising 0.5 M $Li_2S_8$, 1 M lithium bis(trifluoromethane)sulfonimide (LiTFSI), and 0.4 M $LiNO_3$ in diethylene glycol dimethyl ether (2 G). Having 0.5 M $Li_2S_8$ corresponds to an electrolyte-to-sulfur (E/S) ratio of 7.8 μl $mg_S^{-1}$ (Supplementary Note 1). The separator as catholyte reservoir is oversized to ensure that the material deposition on the cathode is not limited by the S amount. The S amount in the catholyte corresponds to a theoretical cathode mass loading of 19.95 mg cm$^{-2}$ (Supplementary Note 1). Equivalent operando SANS experiments are conducted with a similar custom-built operando cell (Supplementary Fig. 1). Electrode and separator dimension for the SANS measurements are slightly different with a theoretical cathode mass loading of 19.29 mg cm$^{-2}$. A deuterated 2 G solvent improves materials contrast

and minimizes the carbon scattering contribution with SANS. To verify that our findings hold more generally, we also perform operando SAXS/WAXS experiments on another electrochemical energy storage system with a carbon black / sulfur composite cathode and 1 M LiTFSI in tetraethylene glycol dimethyl ether:dioxolane (TEGDME:DOL, 1:1) without PSs as the electrolyte (Supplementary Fig. 2). SAXS and WAXS intensities are recorded on separate areal detectors (Fig. 1a) with a time resolution of 1 min during potentiostatic discharge/charge. The X-ray beam hit the Li metal anode, the catholyte-soaked separator and the carbon black cathode. All reversible structural changes seen by operando SAXS/WAXS and SANS stem from the reversible deposition/dissolution of active material in the carbon cathode only (Supplementary Figs. 3, 4). More details are given in the Methods.

The (dis)charge profile in the operando cell shows the expected behavior of a Li-S electrochemical energy storage system (Fig. 1b). The absolute current during potentiostatic discharge at 2.0 V vs. $Li/Li^+$ exhibits a distinct minimum indicating the point where $Li_2S$ formation dominates. After this minimum, the current (i.e., the $Li_2S$ formation rate) increases, since the growth of $Li_2S$ on existing $Li_2S$ nuclei occurs at a higher rate than initial nucleation. The reduction in current after ~4500 s indicates the onset of capacity-limiting processes. The discharge is stopped after 2.5 h at a capacity of 1520 mAh $g_C^{-1}$ (normalized by the carbon mass, as there is no defined amount of sulfur present at the cathode). The maximum theoretical capacity of $Li_2S_8$ in the 60 μl catholyte corresponds to ~18000 mAh $g_C^{-1}$, indicating that the capacity is not limited by the amount of S in the catholyte. Consistent with literature[15,16], ex situ SEM micrographs of electrodes after full potentiostatic discharge show large structures with particle sizes beyond 100 nm (Fig. 1c). Due to the poor electronic conductivity of $Li_2S$ ($>10^{-19} S \, cm^{-1}$ according to Ref. 44.), the resolution of SEM is not sufficient to resolve the nanostructure below 100 nm properly; however, these insights can be obtained by SAXS and SANS. During charge at 2.45 V vs. $Li/Li^+$ for the same time (2.5 h), initially high currents fade quickly after ~2/3 of the capacity (~1 mAh $g_C^{-1}$).

The initial SAXS intensity prior to discharge shows a roughly linear decay in the double-logarithmic plot (Fig. 2a). Such power law behavior is typical for the fractal-like structure of paracrystalline carbon-based electrodes. During discharge, the SAXS intensity generally increases, with a larger increase at high scattering vector length ($q$) around 1.5 nm$^{-1}$ and at low-$q$ around 0.2 nm$^{-1}$. The background-corrected WAXS intensities indicate the formation of $Li_2S$ crystallites during discharge (Fig. 2b).

To visualize the subtle SAXS intensity changes during the full potentiostatic discharge/charge cycle, we plot the relative SAXS intensity change with respect to the initial SAXS intensity prior to discharge as a function of time and scattering vector length $q$ (Fig. 2c, d). The WAXS intensity is also plotted as a function of time and scattering angle in Fig. 2e. As solid $Li_2S$ starts to form (as evidenced by the decreasing current at ~5000 s in Fig. 2c and the emergence of the $Li_2S$ crystallites in Fig. 2e), two distinct maxima appear on the relative SAXS intensity at low $q$ (regime $q_A$) and at high $q$ (regime $q_B$). In line with the high currents during charge (Figs. 1b, 2c), these features disappear quickly during charge compared to their emergence during discharge.

Comparing the changes in intensities of the SAXS and WAXS features (Fig. 2f) shows similarities in the emergence of the WAXS and the high $q$ SAXS feature during discharge. Meanwhile the low $q$ SAXS feature decreases at the end of discharge. During charge, the low-$q$ SAXS feature decreases quickest. The WAXS signal from the $Li_2S$ crystallites decreases more slowly, with the high-$q$ feature decreasing even slower. These observations suggest that the relative SAXS intensity maxima, while related to $Li_2S$ deposition and dissolution, do not correlate directly to the $Li_2S$ crystallites probed by WAXS. The two distinct maxima may be caused by more than one solid discharge product.

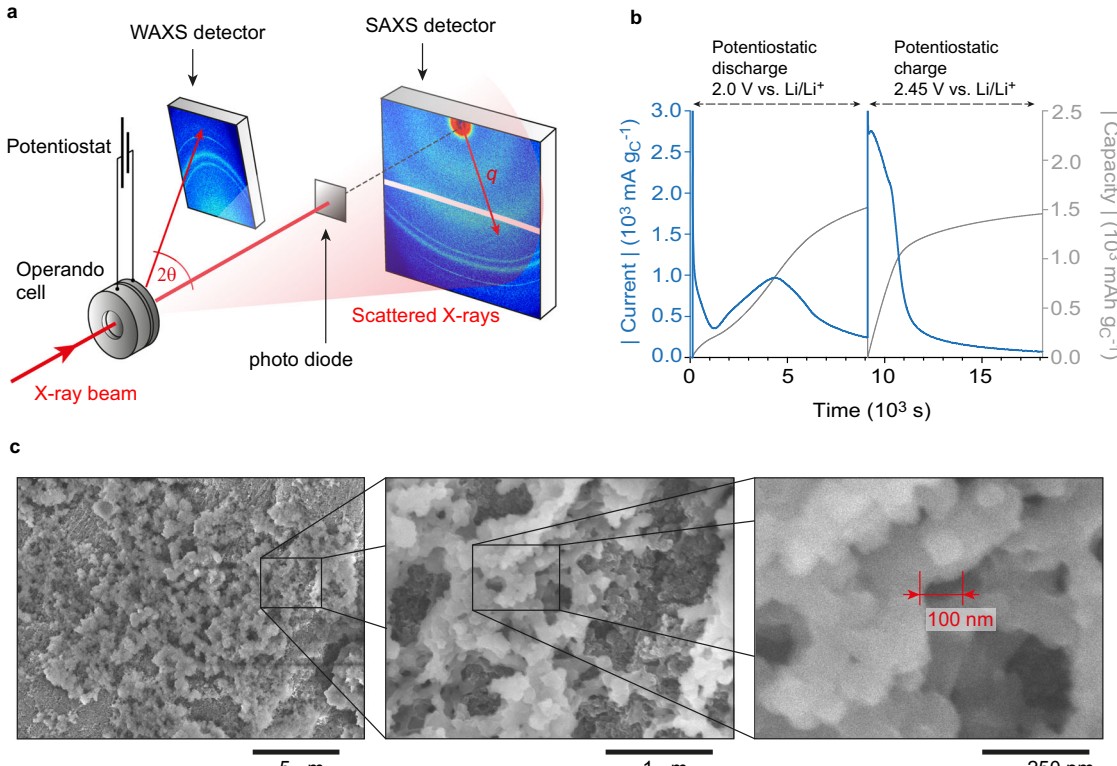

**Fig. 1 | Operando and ex situ scattering, electrochemical and microscopy measurements. a** Sketch of the experimental set-up for operando SAXS / WAXS measurements carried out at the international research centre ELETTRA[62] showing the separate detectors. **b** Absolute specific current (blue) and absolute specific capacity (grey) versus time during potentiostatic discharge/charge of the operando SAXS cell at 2.0 V / 2.45 V vs. Li/Li[+] at a temperature of 25 °C ± 3 °C. Both current and capacity are normalized by the bare carbon black electrode mass. **c** Ex situ scanning electron microscopy images at different magnification show the hierarchical structure of Li$_2$S deposits on the carbon black electrode after potentiostatic discharge at 2.0 V vs. Li/Li[+] to a capacity of 1520 mAh g$_C^{-1}$.

Importantly, the shape of the SAXS curves does not only depend on the amount and morphology of solid particles such as Li$_2$S. The exact SAXS intensity changes are a complex function of all contributing structures (deposit, carbon black and electrolyte) and their cross-correlations (see Eqs. 4–5 for a three-phase system in the Methods section). The decrease of the SAXS intensity in the $q_A$ regime (grey line in Fig. 2f) at the end of discharge, for example, could be explained by a slight increase of the dissolved PS concentration in the electrolyte.

SAXS features arising from solid non-Li$_2$S phases is further supported when considering the sizes of the features. From WAXS, we use the Scherrer equation (details are given in the Methods section) to estimate that the Li$_2$S crystallite size (i.e., mean diameter) increases and plateaus at about ~7 nm (Fig. 2g). A Williamson-Hall analysis[45] on a galvanostatically discharged carbon/S electrode reveals that isotropic strain contributes significantly to peak broadening. It results in a crystal size of about 12.8 nm, compared to 8.9 nm obtained from the Scherrer equation (Supplementary Fig. 5). Spherical ~10 nm single crystal particles should, in a first approximation, cause a broad SAXS intensity shoulder around 0.5 nm$^{-1}$ (Supplementary Fig. 6). However, neither the high-$q$ (1.5 nm$^{-1}$) nor the low-$q$ (0.2 nm$^{-1}$) relative SAXS intensity maximum relates to this primary Li$_2$S crystallite size, instead indicating features of approximately 2.8 nm and 26 nm, respectively (Supplementary Fig. 6).

To verify whether the features seen in the SAXS/WAXS data are specific to our selected materials and operating conditions, we galvanostatically discharge a sulfur / carbon-based electrode in a 1 M LiTFSI / TEGDME:DOL (1:1 vol.%) electrolyte solution at three different currents (Supplementary Fig. 2). For all currents, we find a 6 – 7 nm Li$_2$S crystallite size from the WAXS diffraction peak fitting (Scherrer) and a high $q$ relative SAXS intensity maximum between 1 and 2 nm$^{-1}$. Primary Li$_2$S crystallite formation can therefore not be explained by classical

nucleation and continuous growth[46,47], which would result in a crystallite size that strongly depends on current.

On the other hand, the low $q$ intensity maximum depends on the applied current (Supplementary Fig. 2). With increasing current, the intensity shifts to higher $q$-values (from ~0.1 nm$^{-1}$ at 1.23 mA cm$^{-2}$ to «0.08 nm$^{-1}$ at 0.12 mA cm$^{-2}$). We therefore attribute our low $q$ feature to aggregates comprising of smaller primary Li$_2$S crystallites. At higher current, we have more, smaller aggregates, which is in principle consistent with heterogenous nucleation and growth[48].

These SAXS/WAXS findings are in line with experimental data previously discussed in the literature. Independently of the used electrode materials, electrolyte solutions or applied current[13,15,16,26,42,48], the Li$_2$S primary crystallite size has been shown to remain around 10 nm. Size and shape of the super-structures (aggregates) on the other hand, are very sensitive to the used electrolyte and conditions such as current density[11,15,16,48]. A feature similar to our signal at low-$q$ was also observed using small angle neutron scattering[41,42]. Finally, the Li$_2$S deposits observed via ex situ SEM measurements are known to be larger than the primary crystallite size estimated by XRD measurements (in situ and ex situ) via the Scherrer equation or a Williamson-Hall plot[15,16].

New in this work is the identification of the high-$q$ SAXS intensity maximum corresponding to a feature with ~2.8 nm diameter, which is not Li$_2$S. Understanding the origin of this feature can provide the missing piece of the puzzle in quantifying Li$_2$S formation and dissolution.

## Physicochemical investigations on the solid Li$_2$S$_x$ (2 ≤ x ≤ 4) precipitates

The size of Li$_2$S crystallites of ~10 nm cannot explain the high-$q$ SAXS intensity shoulder around 1.5 nm$^{-1}$. The high-$q$ feature disappears when washing (with 2 G) under inert conditions (Argon atmosphere)

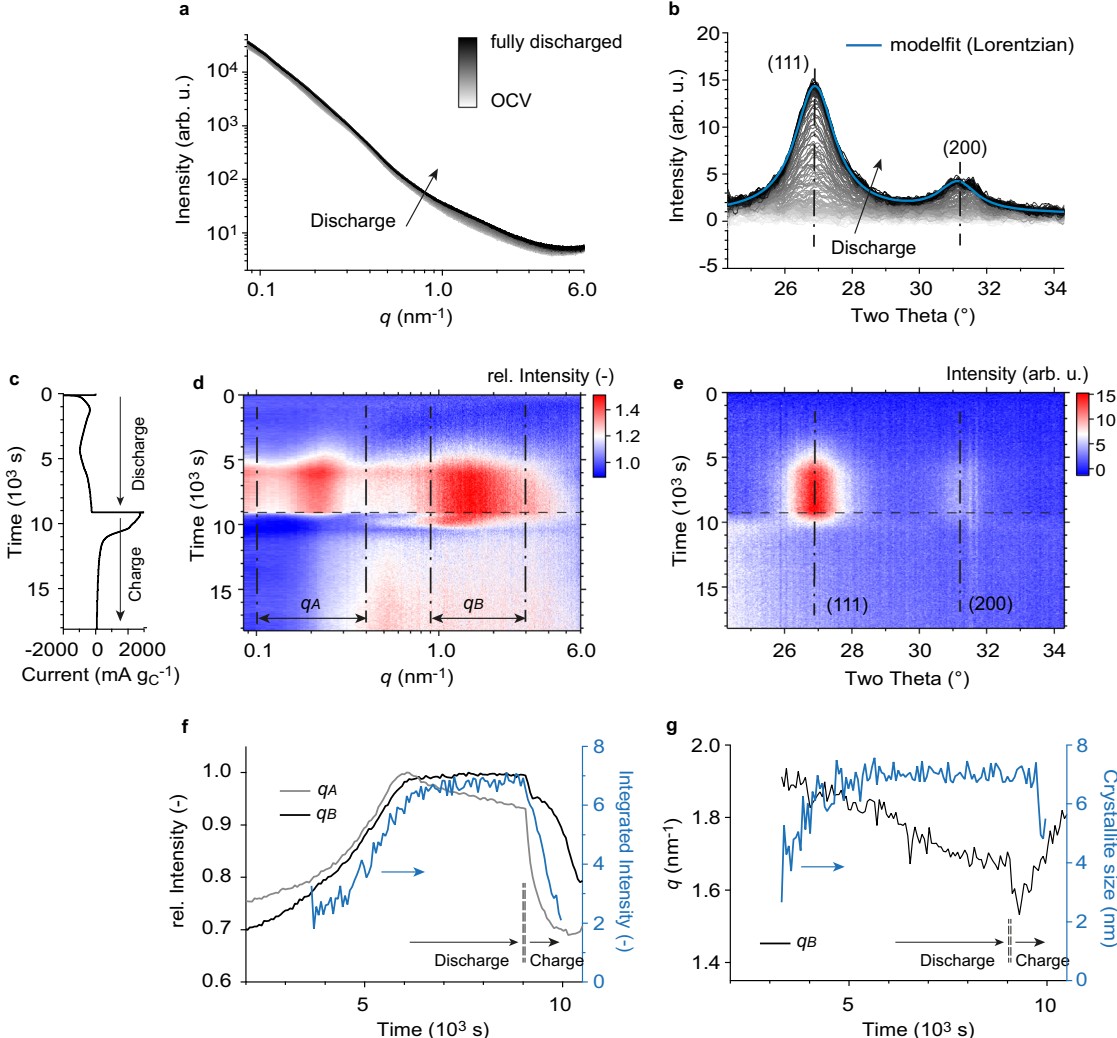

**Fig. 2 | Operando SAXS/WAXS measurements. a** SAXS intensities versus scattering vector length $q$ during potentiostatic discharge at 2.0 V vs. Li/Li$^+$ up to a capacity of 1520 mAh g$_C^{-1}$. **b** The respective background corrected WAXS intensities versus scattering angle during potentiostatic discharge. The (111) and (200) Li$_2$S diffraction peaks are fitted using a Lorentz function. **c** Specific current versus time during potentiostatic discharge/charge at 2.0 V / 2.45 V vs. Li/Li$^+$. **d** The relative SAXS intensity change as a function time and the scattering vector length $q$. The SAXS intensities were normalized by the SAXS intensity prior to discharge at OCV. The $q$-regions $q_A$ and $q_B$ embrace intensity maxima that appear upon Li$_2$S formation

at low and high $q$, respectively. **e** The WAXS intensities as a function of time and scattering angle. The dash-dotted lines indicate the (111) and (200) Li$_2$S diffraction peaks. **f** Normalized, mean SAXS intensity of the low-$q$ ($q_A$) and high-$q$ ($q_B$) regimes during potentiostatic discharge/charge (black and grey) and (111) diffraction peak integrated intensity in blue (obtained from Lorentzian peak fit). **g** Shift of the SAXS intensity maximum in $q_B$, and Li$_2$S crystallite size in blue (obtained from the (111) peak width and the Scherrer equation) during potentiostatic discharge/charge. Details on the quantification of the SAXS $q$-shift and the crystallite size are given in the Methods section.

and drying the discharged electrode under vacuum; the Li$_2$S diffraction peak remains (Fig. 3a, b). Without washing, both the high-$q$ SAXS shoulder and the Li$_2$S diffraction peak remains. This suggests that the high-$q$ SAXS feature is caused by a polysulfide structure that is partially soluble and can be washed away. Li$_2$S$_2$ powder formed by drying a solution with nominal Li$_2$S$_2$ stoichiometry in tetrahydrofuran (THF) solvent shows the same SAXS feature at a similar $q$-position. Ex situ Raman spectroscopy measurements of a potentiostatically discharged and washed (2 G) glassy carbon electrode indicates Li$_2$S, but also a short-chain PS (Li$_2$S$_2$, Li$_2$S$_3$ or Li$_2$S$_4$) and less Li$_2$S$_6$/S$_3^-$ (Fig. 3c)[23,49]. Instead of the carbon black, we used a glassy carbon electrode because of the lower absorbance and a better signal-to-noise ratio. In Supplementary Fig. 7 we show reference spectra of S, Li$_2$S and solid PSs with nominal Li$_2$S$_2$ and Li$_2$S$_4$ stoichiometry. Nominal Li$_2$S$_2$ (Supplementary Fig. 7) shows similar peaks like the discharged electrode at 373 cm$^{-1}$ (Li$_2$S), 440 cm$^{-1}$ (Li$_2$S$_2$) and 534 cm$^{-1}$ (S$_3^-$, Li$_2$S$_6$)[23,49]; the 440 cm$^{-1}$ peak of the

discharged electrode is however too broad to unequivocally assign it to Li$_2$S$_2$.

Amorphous Li$_2$S causing the high SAXS intensity shoulder can be excluded. First, we could not wash it away, as shown in Fig. 3a. Second, the $q_B$ intensity should drop immediately during charge, similar to the Li$_2$S (220) diffraction peak (Fig. 2f).

We conclude that the high-$q$ SAXS shoulder originates from Li$_2$S$_x$ ($2 \le x \le 4$) nanoparticles with a mean diameter ~2.8 nm. Considering the small particle size and the potential isotropic strain, the Li$_2$S$_x$ WAXS diffraction peaks are broad and indistinguishable from the background. The small size and isotropic strain might also explain the large peak widths in the Raman spectra[50]. S$_3^-$ (or Li$_2$S$_6$ in its associated form) could stem from Li$_2$S$_4$ disproportionation ($2S_4^{2-} \rightarrow S_2^{2-} + S_3^-$, or $2Li_2S_4 \rightarrow Li_2S_2 + Li_2S_6$, in the associated form). The same disproportionation to Li$_2$S$_2$ and S$_3^-$ is indeed found in a solution of Li$_2$S$_4$ in 2 G and in previous works[49,51] (Fig. 3c).

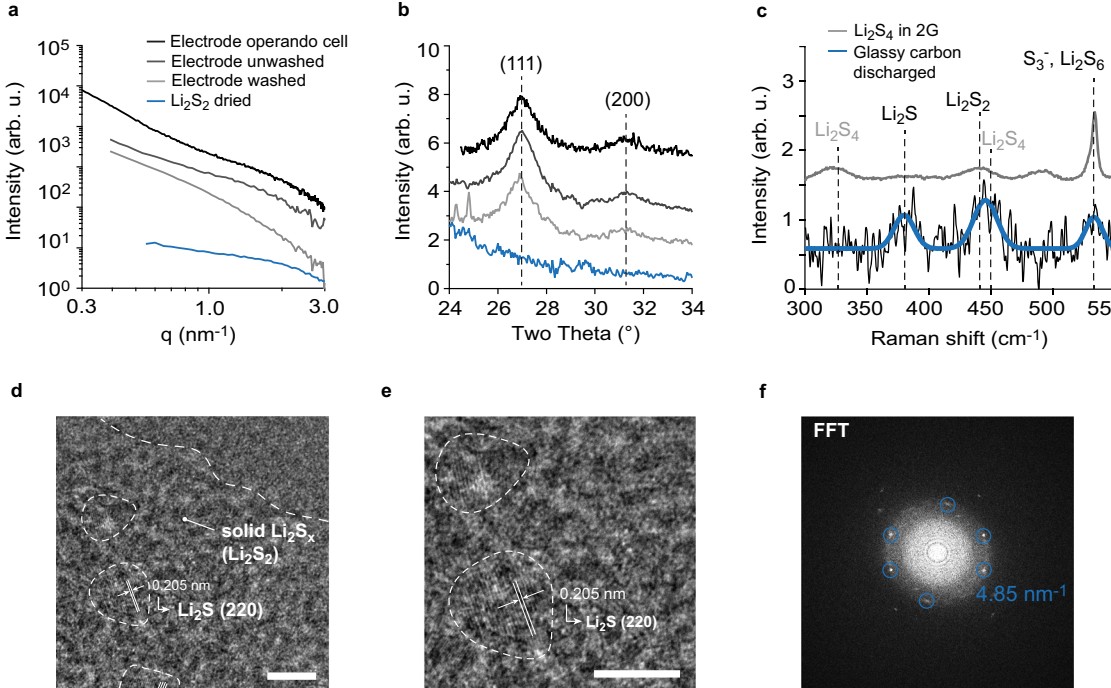

**Fig. 3 | Physicochemical investigation on the presence of solid $Li_2S_x$ ($2 \leq x \leq 4$) particles.** SAXS/WAXS intensities versus scattering vector length $q$ (**a**) and scattering angle (**b**) for the discharged positive electrode in the operando cell. The black solid line shows the equivalent to the reduced SAXS intensity after discharge in Fig. 5a, the discharged positive electrode without washing, but with drying under vacuum (dark grey solid line) and the discharged positive electrode after washing with diglyme (2G) and subsequent drying under vacuum (grey grey solid line). The blue solid line shows the SAXS/WAXS intensities of dissolved $Li_2S_2$ in a solution of tetrahydrofuran (THF) and dried under vacuum. **c** Raman intensities versus Raman shift for a potentiostatically discharged glassy carbon electrode (black solid line).

The blue solid line show Gaussian peak fits, proving the presence of $Li_2S$, $Li_2S_2$ and $S_3^-$ ($Li_2S_6$)[23,49]. The grey solid line corresponds to a solution of dissolved $Li_2S_4$ in 2 G, showing the disproportionation of $Li_2S_4$ into $Li_2S_2$ and $S_3^-$ ($Li_2S_6$). These measurements indicate that the high-$q$ SAXS intensity shoulder -1.5 nm$^{-1}$ is caused by solid short-chain PS particles. **d** TEM image of a galvanostatically discharged C/Au grid using a $Li_2S_4$ catholyte. Scalebar 5 nm. **e** The same image with higher magnification. The lattice fringes in the marked areas have a d-spacing of 0.205 nm, which fits to the $Li_2S$ (220) lattice planes. Scalebar 5 nm. **f** The corresponding Fast Fourier Transformation (FFT).

TEM measurements of $Li_2S/Li_2S_x$ films electrodeposited on a C/Au TEM grid, gives further evidence for the co-existence of the two solid discharge products: nanocrystalline $Li_2S$ and nanoparticulate, amorphous $Li_2S_x$ (Fig. 3d–f). The TEM images show ~6 nm large $Li_2S$ crystals, embedded in a matrix of amorphous material, which is likely $Li_2S_x$. The fast Fourier transformation (FFT) image indicates peak positions that fit to a $d$-spacing of 0.205 nm. This $d$-spacing can be assigned to the (220) plane in $Li_2S$.

## Development of a structural model to interpret SAXS and SANS intensities

Our experiments (SAXS/WAXS, SEM, TEM, Raman) suggest that discharging a Li-S battery positive electrode results in a composite structure consisting of solid $Li_2S$ and short-chain $Li_2S_x$ particles ($2 \leq x \leq 4$). The solid $Li_2S_x$ particles are responsible for the SAXS feature in region $q_B$ in Fig. 2d and have a mean size around 2.8 nm. The ~10 nm $Li_2S$ crystallites (as shown in the WAXS measurements and analyses) aggregate to form features with a mean size (diameter) around 26 nm (region $q_A$ SAXS). These polycrystalline aggregates arrange into the larger structures >100 nm as shown in the SEM micrographs (Fig. 1c). During charge, the aggregates first dissolve into primary $Li_2S$ and $Li_2S_x$ particles. $Li_2S_x$ dissolution is lagging behind $Li_2S$ dissolution during the entire charge (in Fig. 2f, the low-$q$ maximum disappears faster than WAXS diffraction peaks and high-$q$ shoulder).

Given the low solubility of $Li_2S_2$[52], previous studies have speculated whether solid $Li_2S_2$ is present as a second discharge product in Li-S batteries, but so far it has not clearly been observed experimentally[22–25]. Discussions about the existence of $Li_2S_2$ are mostly

based on electrochemical data, its stability predicted by DFT or the fact that operando absorption spectroscopy typically finds a mixture of $Li_2S$ and dissolved short-chain PSs at the end of discharge[23,28,53,54]. However, a direct structural or spectroscopic evidence for $Li_2S_2$ as a second, solid discharge product is still missing[55]. The low solubility of $Li_2S_2$, the similarity between $Li_2S_2$ reference spectrum (Supplementary Fig. 7) and discharged electrode (Fig. 3c), and the disproportionation of $Li_2S_4$ to form $Li_2S_2$ (see Fig. 3c and Ref. 51.) all suggest that our observed solid short-chain PS phase is $Li_2S_2$. To validate this and gain further quantitative insights into the structural evolution of the $Li_2S/Li_2S_2$ nanostructure, we now analyze equivalent operando SANS experiments during potentiostatic discharge/charge using a deuterated 2 G solvent. Compared to SAXS, the materials contrast of $Li_2S$ and $Li_2S_2$ is improved with SANS (Fig. 4a, Supplementary Table 1). Further, the deuterated catholyte (0.5 M $Li_2S_8$ + 1 M LiTFSI + 0.4 M $LiNO_3$ in deuterated 2 G) minimizes the carbon scattering contribution due to similar scattering length density (SLD) of carbon and catholyte (see Supplementary Note 2 and Supplementary Fig. 8). To avoid the contribution of the electrolyte structure factor and the incoherent background, we subtract the SANS contribution prior to discharge from all operando data obtained (see the Methods section for further details). The background-corrected operando SANS intensities during potentiostatic discharge/charge indeed show large intensity changes, entirely attributed to $Li_2S/Li_2S_2$ formation (Fig. 4b, c).

The concept of plurigaussian random fields[40,56] (PGRF) is used to fit the operando SANS data and to create a stochastically representative three-phase $Li_2S/Li_2S_2$/electrolyte structure in real-space (see the Methods section for further details). By fitting the SANS intensities

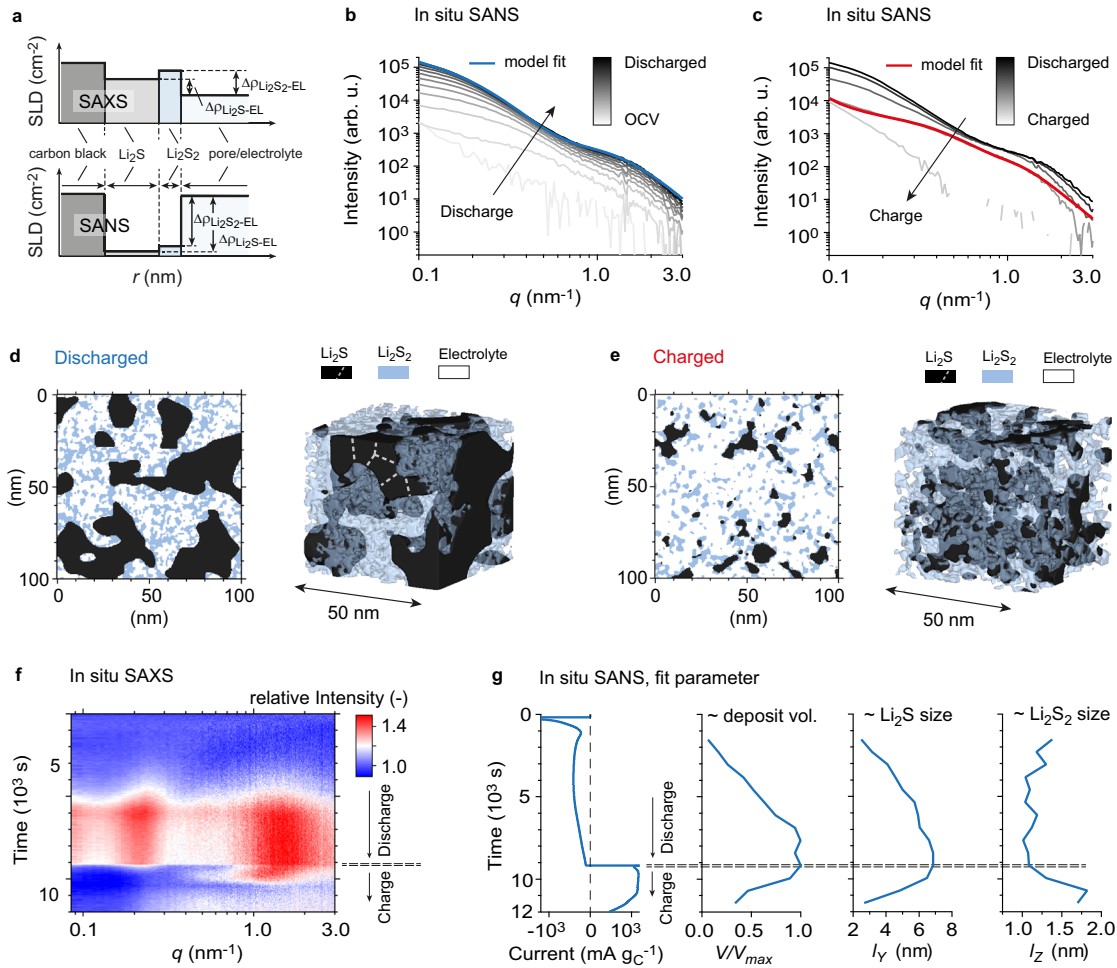

**Fig. 4 | A stochastic SANS model to describe the nanoscale phase evolution of Li₂S / Li₂S₂ deposits. a** Sketch of scattering length densities of the different phases during a SAXS experiment (top) and a SANS experiment (bottom). The difference of scattering length densities (SLDs) is related to the materials contrast during a scattering experiment. SANS has higher materials contrast than SAXS and minimizes the carbon scattering contribution. Detailed numbers are given in Supplementary Table 1. **b** Background-corrected SANS intensities versus scattering vector length $q$ during potentiostatic discharge at 2.0 V vs. Li/Li⁺. The plurigaussian random field (PGRF) model fit at the end of discharge is given in blue. **c** Background-corrected SANS intensities versus scattering vector length $q$ during potentiostatic charge at 2.45 V vs. Li/Li⁺. The plurigaussian random field (PGRF) model fit after 38 min of charging is given in red. **d**–**e** The corresponding representative real space models generated via PGRFs and the input parameters obtained from the model fits in **b** and **c**. **d**–**e** show a 100 nm² cross section, and a 3D visualization of a 50 nm³ cut-out. White dashed lines indicate schematically that Li₂S aggregates consist of

individual Li₂S grains. **f** Experimental relative SAXS intensity change as a function of time and scattering vector q during potentiostatic discharge/charge. The data is the same as in Fig. 2d and shown for direct comparison with SANS fits. **g** Current signal and fit parameters as a function of time, obtained from PGRF model fits of the operando SANS data in **b** and **c**. The parameter $V/V_{max}$ corresponds to the relative amount of Li₂S/Li₂S₂ deposit during the experiment. $l_Y$ and $l_Z$ correspond to a correlation length of the Li₂S and Li₂S₂ deposits, respectively. These length values are proportional to the actual particle sizes, which can be seen in **d** and **e**. Based on the $q$-position the particle/aggregate diameters were estimated to be around 26 nm and 2.8 nm at the end of discharge (see Supplementary Fig. 6). The Li₂S aggregate size clearly increases during discharge, the Li₂S₂ particle size slightly decreases during discharge, before it increases at the beginning of charge. The shift of the high-$q$ SAXS intensity maximum during charge (**f**) shows the same behavior with better time resolution.

during potentiostatic discharge at 2.0 V (Fig. 4b) an during potentiostatic charge at 2.45 V (Fig. 4c), we extract parameters for (i) the feature sizes of Li₂S and Li₂S₂, (ii) the respective volume fractions of Li₂S and Li₂S₂, and (iii) a parameter accounting for the spatial correlation between the Li₂S and Li₂S₂ structures (Supplementary Table 2). All SANS model fits are given in Supplementary Fig. 9. The parameter ($\delta$, see Methods and Supplementary Fig. 10) defines whether Li₂S₂ particles are preferably located close to the Li₂S surface ($\delta \to 0°$) or randomly distributed across cavities that form amongst the Li₂S particles ($\delta \to 90°$). The value of $\delta = 70°$ shows that Li₂S₂ particles are distributed nearly randomly across the cavities, only with a slight preference to occur in proximity to the Li₂S crystallites. With these parameters, we generate a 3D representation of the Li₂S/Li₂S₂ nanostructure on a 3D lattice after full discharge (Fig. 4d, corresponding to model fit in Fig. 4b) and during charge (Fig. 4e, corresponding to model fit in

Fig. 4c). These visualizations highlight the smaller size of Li₂S₂ particles compared to Li₂S particles, the nearly uniformly distributed Li₂S₂ across the Li₂S cavities, and the mean aggregate size of ~26 nm at the end of discharge. During charge the Li₂S aggregate size decreases steadily, while the Li₂S₂ particle size increases. Similar results have been obtained from a PGRF model fit using operando SAXS (Supplementary Fig. 11). Due to the non-negligible carbon scattering contribution in carbon black electrodes, the Li₂S/Li₂S₂ model fits for SAXS were carried out using electrodes with larger glassy carbon beads with an otherwise same cell configuration. The large size of the glassy carbon beads (>5 μm) ensures that their SAXS scattering contribution is negligible.

While operando SANS has advantages in terms of contrast, operando SAXS has a higher time resolution. In Fig. 4f, g, we compare the relative SAXS intensity changes and the operando SANS fit

parameters as a function of time. Fig. 4g shows the normalized $Li_2S$/$Li_2S_2$ deposit volume as obtained from the time-dependent PGRF model fits. The deposit volume grows fast at initial stages of discharge before it reaches a plateau, and decreases during charge. During initial states of charge, $Li_2S_2$ particles grow (see $l_Z$, Fig. 4g), while the $Li_2S$ aggregates dissolve into their primary crystals (see decrease of $l_Y$, Fig. 4g). This explains the high-$q$ maximum shifting to smaller $q$ in the relative SAXS intensity plot (Fig. 4f). $Li_2S_2$ dissolution lags behind $Li_2S$ dissolution during the entire charge, shifting relative volume fractions towards $Li_2S_2$. All operando SANS fit parameters are shown in Supplementary Fig. 12.

In summary, the SAXS/WAXS data in Fig. 2, Supplementary Fig. 2, Supplementary Fig. 11, the SANS data in Fig. 4, and the TEM micrographs in Fig. 3 all suggest the presence of nanocrystalline $Li_2S$ and a second solid discharge product, such as $Li_2S_2$. These findings are therefore valid for a broad range of E/S ratios, S mass loadings, applied currents and carbon hosts (experimental parameters of all investigated systems are summarized in Supplementary Table 3).

## Physicochemical investigations on the $Li_2S$ formation via $Li_2S_2$ precipitation and solid-state conversion

The found $Li_2S$/$Li_2S_2$ composite structure is not compatible with a simple step-wise electroreduction of polysulfides at the carbon|electrolyte interface. Instead, the structural features point to species in the electrolyte supporting growth. This could be $Li_2S$ if it redissolves ($Li^+$, $S^{2-}$) and precipitates after having formed by direct reduction at the carbon; however, the low solubility of $Li_2S$[15] suggests that dissolved $Li^+$ and $S_2^-$ could only form small $Li_2S$ crystallites (<10 nm) on or in close proximity to the carbon surface, leading quickly to a passivating surface film[21]. Alternatively, the aggregate superstructures could be formed by precipitation of $Li_2S_2$, which has a higher solubility than $Li_2S$. $Li_2S$ could then be formed via solid-state electroreduction. The latter requires sufficiently facile ambipolar transport ($Li^+$ and $e^-$) in the solid state.

We next investigate whether the $Li_2S_2$ reduction to form $Li_2S$ can in principle occur in the solid state and fast enough to occur during battery discharge. We rolled crystalline solid sulfur onto a piece of Li metal (without any liquid electrolyte added) in a molar ratio of 1:2 under Ar atmosphere and recorded the XRD pattern of the resulting mixture from 3 to 20 h after mixing (Fig. 5a, details see Methods). The crystallite size obtained from the diffraction peak widths remained relatively constant with increasing reaction time and similar to the $Li_2S$ size obtained from electrochemical discharge (Fig. 5b). This suggests that in Li-S batteries with non-aqueous liquid electrolytes, the $Li_2S$ during cell discharge is formed by a similar solid-state conversion process as seen in Fig. 5. The experiment further shows that solid S can

convert into $Li_2S$ within a few hours only, despite mean Li diffusion pathways of more than 50 μm. With the same conversion rate, a 1 μm thick S film could be converted to $Li_2S$ in less than an hour. This is significantly faster than the ionic and electronic conductivities of $Li_2S$ suggest[57]. Transport at the nanocrystal interfaces and grain boundaries might be enhanced significantly.

Ex situ Raman spectroscopy measurements (Fig. 3c) indicates the tendency of $Li_2S_4$ ($S_4^{2-}$) to disproportionate into $Li_2S_2$ ($S_2^-$) and $Li_2S_6$ ($S_3^-$). $Li_2S$ formation in Li-S batteries can thus be explained as a combination of $Li_2S_2$ precipitation from solution via $Li_2S_4$ disproportionation (and/or $Li_2S_4$ electroreduction) and a solid-state reduction to $Li_2S$ (Eq. 1).

$$4/3 Li_2S_{6(sol)} + 4/3 Li^+ + 4/3 e^- \rightleftharpoons 2 Li_2S_{4(s)} \overset{DISP}{\underset{COMP}{\rightleftharpoons}} Li_2S_{6(sol)} + Li_2S_{2(s/sol)}$$
$$Li_2S_{2(s)} + 2 Li^+ + 2 e^- \rightleftharpoons 2 Li_2S_{(s)}, \tag{1}$$

with DISP meaning disproportionation and COMP meaning comproportionation. The processes in Eq. 1 are illustrated in Fig. 6. Solution-mediated $Li_2S_4$ disproportionation, $Li_2S_2$ precipitation, and subsequent solid-state reduction explain why $Li_2S$ deposits do not passivate the carbon surface at the positive electrode[16,58], even though the low $Li_2S$ solubilities would imply so[15,21,59]. $Li_2S_2$ precipitation from solution can cause a variety of shapes (e.g. platelets[15,16]) and sizes beyond several 100 nm, where the primary $Li_2S$ crystal size (as observed by XRD measurements) is constantly around 10 nm. We believe that the $Li_2S$ crystallite size is limited because of the large mechanical stress that evolves when $Li_2S$ forms in a solid matrix with higher density, such as $Li_2S_2$. Upon phase transformation, the expansion to $Li_2S$ would be suppressed. Further, $Li_2S_2$ precipitation from solution explains why $Li_2S$/$Li_2S_2$ aggregate size and shape depend strongly on current[48] and solvent[15], while the $Li_2S$ primary crystallite size does not.

Solid-state electroreduction from $Li_2S_2$ to $Li_2S$ requires fast enough ambipolar $Li^+$ and $e^-$ solid-state transport. Theoretical works suggest that solid polysulfides such as $Li_2S_2$ have indeed slightly higher electronic conductivities compared to $Li_2S$ or S[55,60]. For a specific $Li_2S_2$ nanostructure, the high surface area and the richness of defects further increase the diffusivity compared to bulk crystalline $Li_2S_2$[61]. We speculate that during solid-state conversion $Li^+$ ions diffuse through the porous deposits to the carbon|electrolyte interface. The $Li_2S_2$ reduction takes place at triple-phase boundaries ($Li_2S_2$, carbon, electrolyte). Sufficiently fast chemical diffusion ($Li^0$) via the $Li_2S_2$ nanostructure and $Li_2S_2$ surfaces (or grain boundaries) convert the $Li_2S_2$ structure into $Li_2S$.

Charging reverses the processes shown in Fig. 6. While $Li_2S$ aggregates dissolve steadily, solid $Li_2S_{2(s)}$ particles grow at initial

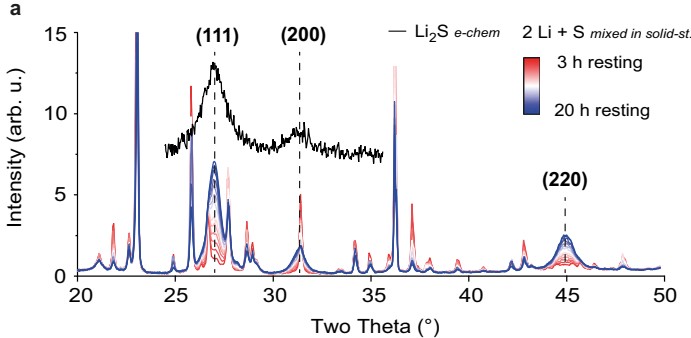
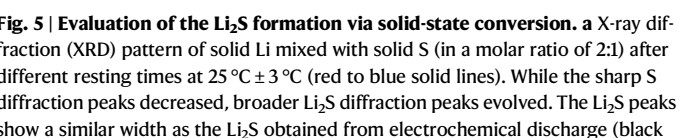
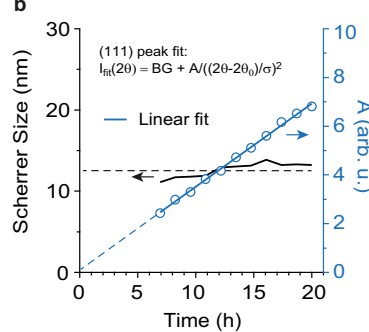

**Fig. 5 | Evaluation of the $Li_2S$ formation via solid-state conversion. a** X-ray diffraction (XRD) pattern of solid Li mixed with solid S (in a molar ratio of 2:1) after different resting times at 25 °C ± 3 °C (red to blue solid lines). While the sharp S diffraction peaks decreased, broader $Li_2S$ diffraction peaks evolved. The $Li_2S$ peaks show a similar width as the $Li_2S$ obtained from electrochemical discharge (black solid line). **b** The Scherrer crystallite size of the (111) peak is 7 nm for $Li_2S$ obtained from electrochemical discharge and around 12.5 nm for $Li_2S$ obtained by mixing solid S and Li. With increasing resting time, the $Li_2S$ diffraction peaks grow, their width and the crystallite size remain constant. This is shown based on a Lorentzian peak fit of the form $I(2\theta) = BG + A/((2\theta - 2\theta_0)/\sigma)^2$.

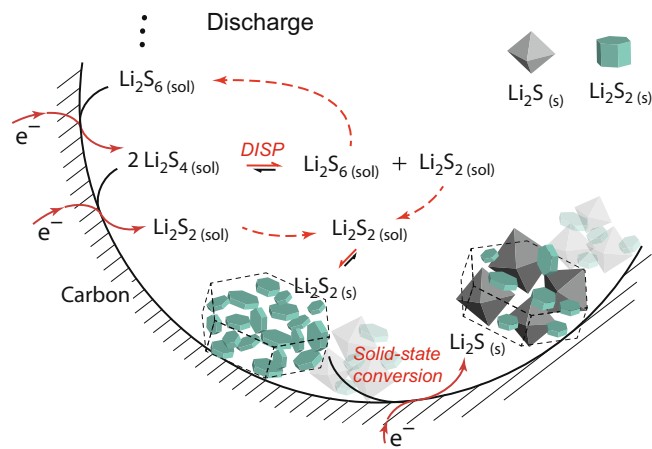

**Fig. 6 | Schematic representation of the proposed Li₂S formation mechanism during the discharge of a Li-S battery with nonaqueous liquid electrolyte. a**, Processes and equilibria upon discharge. The Li₂S/Li₂S₂ aggregate shape is defined by Li₂S₂ precipitation, which is formed by Li₂S₄ electroreduction and/or Li₂S₄ disproportionation (DISP). Li₂S is formed by solid-state conversion from Li₂S₂, likely via solid-state electroreduction. Alternative to electroreduction, Li₂S could be formed via solid-state disproportionation (3Li₂S₂(s) ⇌ 2Li₂S(s) + Li₂S₄(s,sol) and 2Li₂S₄(s,sol) ⇌ Li₂S₆(sol) + Li₂S₂(s/sol)). Dashed arrows indicate diffusion. Solid Li₂S and Li₂S₂ are illustrated according to their Wulff shapes (crystal shape in thermodynamic equilibrium)[60].

stages of charge, as seen in Fig. 4g and qualitatively based on the shift of the high-$q$ SAXS maximum in Fig. 4f. Additional Li₂S₂(s) could be formed from oxidizing solid Li₂S, and due to the increased concentration of dissolved Li₂S₄ which feeds into the disproportionation reaction (Eq. 1) to form Li₂S₂.

Alternative to electroreduction, the final solid-state step during Li-S discharge could also be solid-state disproportionation, for example, via the reaction 3Li₂S₂(s) ⇌ 2Li₂S(s) + Li₂S₄(s,sol). Li₂S₄ would then form solid Li₂S₂ via the disproportionation reaction in Eq. 1, 2Li₂S₄(s,sol) ⇌ Li₂S₆(sol) + Li₂S₂(s/sol), resulting in the composite Li₂S/Li₂S₂ structure shown in Fig. 4.

## Discussion

In conclusion, we provide direct experimental evidence that next to solid Li₂S crystallites, smaller solid short-chain Li₂Sₓ particles are formed upon discharge in non-aqueous Li-S batteries with liquid electrolyte solutions. We demonstrate that these particles are likely Li₂S₂. The particles are small (~2.8 nm), amorphous and thus only visible in SAXS/SANS (the peak broadening is too large to detect them with XRD/WAXS). During charge, the Li₂S₂ particles initially grow while Li₂S disappears. The behavior is consistent with Li₂S₂ precipitation from solution and subsequent solid-state conversion to form Li₂S crystals. Next to electroreduction at the carbon-electrolyte interface, Li₂S₂ is likely formed via disproportionation from Li₂S₄.

Converting Li₂S₂ to Li₂S (i.e. S₂²⁻ → S²⁻) accounts for half of the theoretical capacity of Li-S cells and is – as we propose – a solid-state process. Electroreduction of dissolved Li₂S₂ would lead to a fast coverage of the entire carbon surface at the positive electrode and poor electrochemical energy storage behaviour. Hence, the final solid-state conversion is either a solid-state reduction or solid-state phase separation (disproportionation). There are two arguments speaking for solid-state electroreduction from Li₂S₂. First, solid-state S to Li₂S conversion can be fast (as shown in Fig. 5). Second, the Li₂S crystallite size limited to around 10 nm (independent of electrolyte, current, and formation mechanism) could be explained by Li₂S formation in a solid matrix of denser Li₂S₂, which would mechanically confine and stress the system.

Since the morphology at larger length scales (as seen by SEM measurements) is determined by solution-mediated Li₂S₂ precipitation, the deposits remain porous and guarantee access to the carbon|electrolyte interface[16,58]. This means that discharge capacity of a Li-S battery cathode is limited by mass transport[16,17] rather than electron transport through a passivating surface film[11]. Theoretical sulfur capacities may never be achieved as a certain amount of short-chain PSs (Li₂S₂, Li₂S₄) remains in solution and/or as a second solid phase. Chemical diffusion (of Li⁰, i.e. Li⁺ and e⁻) through the solid-state Li₂S₂ might further decide how much Li₂S₂ can be interconverted to Li₂S. This depends both on solid-state transport and the length of diffusion pathways determined by the Li₂S₂ morphology. The Li₂S/Li₂S₂ structure formation thus defines achievable Li-S capacities.

Given the known relation between electrolyte solvation and Li₂S aggregate morphology[13,15] (i.e., Li₂S nucleation and growth), we believe that solvation energies influence, on the one hand, Li₂S₂ crystallization in terms of nucleation and size/shape, of which the Li₂S/Li₂S₂ deposits form replicas. On the other hand, the electrolyte determines the redissolution and diffusion of the dissolved PSs, which is critical for approaching theoretical capacities. More broadly, the solid-state reduction mechanism from Li₂S₂ to Li₂S indicates solid-state S conversion to be sufficiently facile, despite the poorly conducting nature of Li₂S. This implies that solid-state S-to-Li₂S conversion (SSC) is possible at practical rates if S/Li₂S structures are properly engineered, which is a very important message for all Li-S design strategies seeking to avoid the polysulfide shuttling problem by utilizing SSC, but so far struggled to convert practical S amounts.

Next to electron microscopy, electrochemical methods and Raman spectroscopy, the essential tools for these insights are operando SAXS/WAXS, operando SANS with contrast matching, and advanced data analysis using stochastic modeling. We show that the combination of these techniques offers unique quantitative structural insights into the complex Li₂S/Li₂S₂ composite structure, at length scales hardly accessible to other methods. In this study, SAXS/WAXS and SANS were particularly useful because of the integral structural information, the sensitivity for both crystalline and amorphous solids, and the ability to study the nanoscale structure under practical conditions in an operando cell. The example of Li₂S/Li₂S₂ deposition demonstrates the power of operando SAXS/SANS and stochastic modelling to clarify mechanisms in complex energy materials more generally and that seamless structural information from atomic to nanometer scale holds key to important mechanistic detail.

## Methods

### Materials

As cathode material we used a carbon black (Ketjenblack, EC-600JD, ANR Technologies) with a specific surface area (evaluated via Brunauer-Emmet-Teller method) of 1400 m² g$_C^{-1}$, a primary particle radius of around 34 nm, and a metal impurity content of <30 ppm. The free-standing film electrodes were prepared by mixing carbon with polytetrafluoroethylene (PTFE, 60 mass% suspension in water, Sigma Aldrich) at 90/10 mass ratio with isopropanol (≥ 99.8 %, Sigma Aldrich). The mixing of 10 min was done by hand in a mortar and in air at a constant temperature of 25 °C. The resulting dough-like material was rolled to a 60 ± 10 μm thick film, washed in acetone (≥99.5%, Sigma Aldrich) / H₂O (18 MΩcm) mixture and finally dried at 120 °C under vacuum (10 mbar) overnight. Because of the lower neutron absorption, the positive electrodes for SANS measurement were rolled to a thickness of 180 μm. As catholyte we used a solution of 0.5 M Li₂S₈ + 1 M lithium bis(trifluoromethane)sulfonimide (LiTFSI, 99.95% trace metals basis, Sigma Aldrich) + 0.4 M lithium nitrate (LiNO₃, 99.99% trace metals basis, Sigma Aldrich) in diethylene glycol dimethyl ether (2G, anhydrous, 99.5%, Sigma Aldrich). In Supplementary Fig. 2 we show operando SAXS/WAXS data using a sulfur infiltrated carbon black cathode (ENSACO 350 G, Imerys, specific surface area of 770 m² g$_C^{-1}$,

metallic impurities <10 ppm, sulfur content <150 ppm) with a solution of 1 M LiTFSI + 0.1 M LiNO$_3$ in 1:1 (v:v) 1,3-dioxolane (DOL, anhydrous, 99.8%, Sigma Aldrich) + tetraethylene glycol dimethyl ether (TEGDME, ≥ 99%, Sigma Aldrich) as electrolyte. The ENSACO 350 G carbon/sulfur composite was prepared in a C:S = 1:2 mass ratio by melt infiltration at 155 °C in a tubular quartz tube furnace under Ar atmosphere at 1 bar (Ar flow of 100 ml min$^{-1}$). The positive electrodes were prepared by mixing the carbon/sulfur composite, the polymer binder polyvinylidene fluoride (PVdF, Sigma Aldrich, average M$_W$ 534000), conductive additive carbon black Printex XE2 (Degussa) in a mass ratio of 80:10:10. The mixture was dissolved in N-methyl pyrrolidone (NMP, Aldrich) and ball milled for 30 min at 300 rpm to obtain a homogeneous slurry. The slurry was then cast on a carbon-coated Al foil (Armor, France) with a doctor blade applicator with a wet thickness of 200 μm. The coated slurry was dried at 50 °C overnight. Electrodes with a diameter of 16 mm were punched out the next day and transferred to an argon-filled glovebox. All solvents were used as received and dried under freshly activated Molecular Sieves (type 4 Å) to achieve H$_2$O concentrations <30 ppm. All salts were dried at elevated temperature (90 °C) and reduced pressure (10 mbar).

The Li$_2$S$_8$ powders were synthesized by mixing a stoichiometric amount of elemental sulfur (powder, 99.98% trace metals basis, Sigma Aldrich) and lithium metal as received (110 μm thick Li foil, high purity, FMC Lithium corporation) in excess of dried tetrahydrofuran (THF, anhydrous, ≥99.9%, inhibitor-free, Sigma Aldrich; the THF was dried in a multistep process using Al$_2$O$_3$, molecular sieves, and distillation, after which the water content was measured by Karl Fischer titration (Mettler Toledo, C20) and kept below 2 ppm). The synthesis procedure was conducted in an argon-filled dry box with controlled levels of water and oxygen content (below 0.1 ppm). The mixture was stirred at slightly elevated temperatures (50 °C) until all the reactants dissolved. THF was then removed under reduced pressure (10 mbar) to obtain dry polysulfide powders. The Li$_2$S$_2$ and Li$_2$S$_4$ powders shown in Supplementary Fig. 7 were prepared equivalently, by mixing Li and S in the right stoichiometry. The commercial Li$_2$S in Supplementary Fig. 7 was purchased from Sigma Aldrich (99.98% trace metals basis).

The solid mixture of S and Li in Fig. 5 was prepared by embedding solid S particles in a solid sheet of Li metal in a Li/S molar ratio of 2:1. First, S crystals (99.98% trace metals basis, Sigma Aldrich) were manually crushed with mortar and pestle (Agate stone) for 10 min under dry conditions in an Argon filled glovebox (H$_2$O < 0.1 ppm, O$_2$ < 3.0 ppm), to end up with a fine S powder (particle size ~ 50 μm). Then, the S powder was rolled onto a thin piece of Li metal (≥99.9%, Alfa Aesar, 0.75 mm thickness) using a rolling bar on a glass plate in inert atmosphere (H$_2$O < 0.1 ppm, O$_2$ < 3.0 ppm). The rolling was continued for ~5 min until the S was embedded inside the Li metal sheet and the Li/S piece was turning brittle.

## Experimental

E/S ratios, sulfur mass loadings, and electrode masses for all operando SAXS/WAXS and SANS measurements are summarized in Supplementary Table 3.

Operando SAXS/WAXS and XRD measurements were carried out with a commercial two-electrode electrochemical *operando* scattering cell (BatterycellSAXS, Anton Paar, Austria). We used polytetrafluoroethylene (PTFE) X-ray windows due to their chemical stability and relatively low background in the SAXS regime. The small diameter of the windows (2 mm) ensures a relatively equal pressure distribution across the cell assembly. It consisted of a Li metal anode (≥99.9%, Alfa Aesar, 0.75 mm thickness, 16 mm diameter), a polypropylene separator (Celgard 2400, 25 μm thickness, 41% porosity), a Freudenberg separator (FS 2225E, polyolefin, thickness 150 μm, electrolyte absorption 130 g m$^{-2}$), a carbon black cathode (diameter of 7 mm, the thickness of 60 ± 5 μm), and an Aluminium grid current collector (Type 901 A, the expanded metal company, 0.3 mm thick, 12 mm diameter,

3.18 × 1.81 mm mesh size). The X-ray beam irradiates all cell materials; reversible and significant structural changes are only detected in the cathode. A Biologic SP240 potentiostat/galvanostat was used for electrochemical cycling.

Operando SAXS/WAXS measurements were carried out on the Austrian SAXS beamline at the Synchrotron ELETTRA[62] (Trieste, Italy) using an X-ray wavelength of 0.154 nm and a Pilatus 1 M SAXS and Pilatus 100 K WAXS detector (Dectris, Switzerland) at a temperature of 25 ± 3 °C. During potentiostatic discharge/charge measurements, SAXS and WAXS patterns were collected with 1 s exposure time (to avoid radiation damage) and 60 s period (to avoid large amounts of data). We discharged the cell partially at 2.0 V vs. Li/Li$^+$ for 2.5 h (giving a capacity of 1520 mAh g$_C^{-1}$) and charged it at 2.45 V vs. Li/Li$^+$ for maximum 2.5 h (to reverse the capacity of 1520 mAh g$_C^{-1}$). Operando SAXS data shown in Supplementary Fig. 2 were recorded on a laboratory SAXS/WAXS instrument (SAXSpoint 2.0, Anton Paar, Austria) with an EIGER2 R 1 M area detector (Dectris, Switzerland) and a time resolution of 5 min. All recorded SAXS patterns were azimuthally averaged and normalized by transmission values. The SAXS background intensity was recorded separately for each cell after removing the cathode. The averaged and normalized background intensity was then subtracted from all recorded operando SAXS curves. The azimuthally averaged 2D operando WAXS data were corrected by subtracting the WAXS intensity prior to discharge (at OCV).

The Scherrer crystallite size $\tau$ in Fig. 2g was obtained from a Lorentzian peak fit and using the Scherrer equation $\tau = (K\lambda)/(\beta\cos(\theta))$ with the shape factor $K = 1$, the wavelength $\lambda = 0.154$ nm, $\beta$ the full width half maximum (FWHM) of the peak, and $\theta$ half of the scattering angle. The mean $q_B$ value und thus the $q$-shift in Fig. 2g was obtained by numerical integration in the $q_B$ region: $\langle q_B \rangle = \int q\, I(q)dq / \int I(q)dq$.

Operando SANS measurements during potentiostatic discharge/charge were performed on the D-22 small angle neutron scattering beamline at the ILL neutron source (Grenoble, France) using a wavelength of 0.5 nm, a beam diameter of 10 mm, and two areal detectors with a sample-to-detector distance of 17.6 m and 1.4 m to achieve an overlapping $q$-region[63]. The measurements were conducted at a temperature of 25 ± 3 °C The custom-built two-electrode operando SANS cell has a similar cell design like the SAXS cell, however, the X-ray windows are replaced by 12 mm aluminum windows. The aluminium guarantees low background and uniform pressure across the cell assembly. The cell consisted of a copper foil current collector (≥99.9%, Schlenk Metallfolien), a Li metal anode (≥99.9%, Alfa Aesar, 0.75 mm thickness, 16 mm diameter), a glassfibre separator (Whatman GF/A, 21 mm diameter, 260 μm thickness, 1.6 μm pore size), a carbon black cathode (13 mm in diameter, 180 μm thick), and an aluminum current collector (≥99.5%, Korf). The neutron beam irradiated all the cell materials; reversible and significant structural changes are only detected on the cathode. The recorded 2D detector intensity signal was acimuthally averaged, corrected for sample holder scattering and electronic background, and normalized by transmission values.

Scanning Electron Microscopy (SEM) images were collected with a Hitachi SU-8200 at 5.0 kV acceleration voltage using a secondary electron detector. Ex situ XRD (and SAXS) measurements (Fig. 3a, b, Fig. 5, Supplementary Figs. 3, 5) were carried out on a Rigaku SmartLab 9 kW System, with rotating Cu anode and 2D solid state detector (HyPix-3000 SL). Raman spectroscopy was carried out on a NT-MDT NTEGRA Spectra with a 561 nm laser. The system is equipped with a Newton Andora CCD detector and a diffraction grating of 1800 gr/mm. Discharged positive electrodes for ex situ measurements (SAXS, XRD, Raman) were prepared by opening cell inside our Ar-filled glovebox (H$_2$O < 0.1 ppm, O$_2$ < 3 ppm), washing them with 500 μL of diethylene glycol dimethyl ether (2 G, anhydrous, 99.5%, Sigma Aldrich) and subsequent drying under reduced pressure (10mbar). To ensure air-free transport and ex situ measurements, also the ex situ SAXS/XRD samples were measured inside the SAXS operando cells. Raman

samples were prepared in an Ar filled glovebox ($H_2O$ < 0.1 ppm, $O_2$ < 3 ppm) on a silicon wafer substrate (Si 100 ± 1° orientation, 500 μm thick, p-type Boron doped, 1 – 10 Ωcm), covered with Mylar foil (Spectro-Membrane, ChemPlex Industries, XRF thin-film window, poly-ethylenterephtalat, 3.6 μm thickness, 30 mm diameter), and sealed with an adhesive tape (Tesa).

The TEM sample was prepared by discharging a 5 μL catholyte (0.1 M $Li_2S_4$ + 0.1 M LiTFSI in 2 G) with a constant current of 0.5 μA on a 400 Au mesh pure C film carbon TEM grid (TedPella, 400 mes Au grid, 25 nm pure carbon film) using a polypropylene separator (Celgard 2400, 25 μm thickness, 41% porosity), and a Li metal anode (≥99.9%, Alfa Aesar, 0.75 mm thickness, 16 mm diameter) in our laboratory coin-cell-type cells (uniaxial pressure of 0.7 ± 0.1 MPa). After discharge the grid was washed with 2 G in a glovebox (Argon, $O_2$ and $H_2O$ < 0.1 ppm) to remove residual polysulfide and salts. To ensure air-free transport to the TEM, the grid was placed in a Gatan 648 double tilt vacuum holder. The HR-TEM images (Fig. 3a–c) were captured on an aberration-corrected JEOL Grand ARM (ScopeM, ETH Zürich) operated at 300 kV. The electron dose was kept at a minimum to prevent electron beam induced damage to the sample.

**Operando SANS data modelling via plurigaussian random fields**

The SAXS/SANS intensity of the discharged cathode can be split into three terms,

$$I(q) = I_{\mathrm{Li_2S,Li_2S_2}}(q) + I_C(q) + BG \tag{2}$$

The first term $I_{\mathrm{Li_2S,Li_2S_2}}(q)$ corresponds to the scattering contribution of the $Li_2S$ / $Li_2S_2$ structure, the second term $I_C(q)$ to the scattering contribution of the electrolyte-filled carbon structure and the third background term to the constant (low-$q$) intensity of electrolyte (and carbon) atomic structure factor. Correlations between carbon black and the $Li_2S$ / $Li_2S_2$ structure can generally not be neglected. Only if the carbon structures are much larger than the $Li_2S$/$Li_2S_2$ deposit structure or if the scattering length density contrast between carbon and electrolyte is zero, the carbon contribution is small and can be simply subtracted or neglected. This is done for SAXS experiments using large glassy carbon beads (see Supplementary Fig. 11) and SANS experiments with deuterated electrolyte to match the scattering length density of the carbon (see Supplementary Table 1 and Supplementary Note 2).

To separate the SANS intensity of the $Li_2S$ / $Li_2S_2$ structure we subtract $I_C(q)$ and BG (the incoherent background), i.e., the SANS intensity measured prior to discharge at OCV.

The SANS intensity of the $Li_2S$ / $Li_2S_2$ nanostructure (Fig. 4a) can be written as

$$I_{\mathrm{Li_2S,Li_2S_2}}(q) = K \left( V/V_{\max} \right) \left[ A \, q^{-4} + I_{\mathrm{PGRF}}(q) \right], \tag{3}$$

with $K$ being a constant that depends on instrumental parameters, such as detector efficiency and irradiated sample volume, and $V/V_{\max}$ the relative volume of the deposited $Li_2S$ / $Li_2S_2$ nanostructure. The first power law term stems from the large $Li_2S$ ($Li_2S_2$) agglomerates beyond 100 nm (see SEM images in Fig. 1c. Given their large expansion, the SANS intensity in the measured $q$ range is proportional to $q^{-4}$ (Porod decay). The second term accounts for the $Li_2S$ / $Li_2S_2$ nanostructure in the size regime between 1 to 50 nm and is modelled via plurigaussian random fields, as described further below. The least square error sum is minimized by particle swarm optimization[64] with reasonable parameter constraints.

The reduced operando SANS data $I_{\mathrm{PGRF}}(q)$ is modelled using the concept of plurigaussian random fields (PGRF)[56]. This allows deriving 3D real space models of the solid $Li_2S$ / $Li_2S_2$ nanostructure at different stages of discharge and charge (Fig. 4). A detailed description of the PGRF method is given by Gommes et al.[56].

The SANS intensity $I_{\mathrm{PGRF}}(q)$ is the Fourier transform of the scattering length density (SLD) correlation function $C(r)$

$$I_{\mathrm{PGRF}}(q) = \int_0^\infty C(r) \frac{\sin(qr)}{qr} 4\pi r^2 \mathrm{d}r \tag{4}$$

$C(r)$ for our three-phase system consisting of phases $Li_2S$, $Li_2S_2$, Electrolyte (EL) can be written as

$$\begin{aligned}
C(r) = & \left( \rho_{\mathrm{Li_2S}} - \rho_{\mathrm{Li_2S_2}} \right) \left( \rho_{\mathrm{Li_2S}} - \rho_{\mathrm{EL}} \right) \left[ P_{\mathrm{Li_2SLi_2S}}(r) - \phi_{\mathrm{Li_2S}}{}^2 \right] \\
& + \left( \rho_{\mathrm{Li_2S_2}} - \rho_{\mathrm{Li_2S}} \right) \left( \rho_{\mathrm{Li_2S_2}} - \rho_{\mathrm{EL}} \right) \left[ P_{\mathrm{Li_2S_2Li_2S_2}}(r) - \phi_{\mathrm{Li_2S_2}}{}^2 \right] \\
& + \left( \rho_{\mathrm{EL}} - \rho_{\mathrm{Li_2S}} \right) \left( \rho_{\mathrm{EL}} - \rho_{\mathrm{Li_2S_2}} \right) \left[ P_{\mathrm{ELEL}}(r) - \phi_{\mathrm{EL}}{}^2 \right]
\end{aligned} \tag{5}$$

Here, $\rho_i$ is the scattering length density, $\phi_i$ the volume fraction and $P_{ii}(r)$ the two-point correlation function of phase i.

Using clipped Gaussian random fields, a 3D model of a two-phase nanopore structure can be generated from a fit to the experimental SANS intensity of the structure[65–68]. Plurigaussian random fields combine two Gaussian random fields to model SANS intensities and 3D real space structures of three-phase systems. A Gaussian random field $Y(\mathbf{x})$ is the sum of cosine waves with wave vector lengths distributed according to their power spectral density $f_Y(k)$ and phase factors $\varphi_i$ randomly distributed between 0 and $2\pi$ [55,65,69,70].

$$Y(\mathbf{x}) = \sqrt{\frac{2}{N}} \sum_{i=1}^{N} \cos \left( \mathbf{k}_i \cdot \mathbf{x} - \varphi_i \right) \tag{6}$$

A possible analytic two-point correlation function of the GRF is[68]

$$g_Y(r) = \frac{1}{\cosh(r/l_Y)} \cdot \frac{\sin(2\pi r/d_Y)}{(2\pi r/d_Y)} \tag{7}$$

with $l_Y$ being a correlation parameter related to the mean size of the structure and $d_Y$ a parameter accounting for ordering effects via the second oscillation term. The corresponding power spectral density is

$$f_Y(k) = \frac{k}{\pi} l_Y d_Y \frac{\sinh(\pi k l_Y / 2) \sinh(\pi^2 l_Y / d_Y)}{\cosh(\pi k l_Y) + \cosh(2\pi^2 l_Y / d_Y)} \tag{8}$$

We now define the threshold values $\alpha$ for the Gaussian distributed $Y(\mathbf{x})$ values to generate a two-phase porous structure from the GRF. All spatial coordinates $\mathbf{x}$ with $\alpha < Y(\mathbf{x}) \leq \infty$ are assigned to the pore space (i.e. phase $Li_2S_2$ + EL); all other coordinates to the $Li_2S$ skeleton. The two threshold values are related to the $Li_2S$ volume fraction $\phi_{\mathrm{Li_2S}}$ via:

$$\phi_{\mathrm{Li_2S}} = \frac{1}{\sqrt{2\pi}} \int_\alpha^\infty \exp\left(-\frac{t^2}{2}\right) \mathrm{d}t \tag{9}$$

To model the real space structure and SAXS intensity of the three-phase system, a second independent GRF $Z(\mathbf{x})$ is generated using the same correlation function (Eqs. 7–8) but different input parameters $l_Z$ and $d_Z$ (Supplementary Fig. 10c). The $Li_2S_2$ phase with the volume fraction $\phi_{\mathrm{Li_2S_2}}$ is obtained by cutting $Z(\mathbf{x})$ and $Y(\mathbf{x})$ based on Eq. 10 (and the cut-offs visualized in Supplementary Fig. 10).

$$\phi_{\mathrm{Li_2S_2}} = \iint\limits_{(Y,Z) \, \epsilon \, D_{\mathrm{Li_2S_2}}} \frac{1}{2\pi} \exp\left(-\frac{Y^2 + Z^2}{2}\right) \mathrm{d}Y \mathrm{d}Z \tag{10}$$

The two-point correlation function of the $Li_2S_2$ phase is calculated via

$$P_{Li_2S_2,Li_2S_2} = \int_{D_{Li_2S_2}} dY_1 dZ_1 \int_{D_{Li_2S_2}} dY_2 dZ_2 G_{g_{Y(r)}}(Y_1,Y_2) G_{g_{Z(r)}}(Z_1,Z_2) \quad (11)$$

with $G_g(Y_1,Y_2)$ being the bivariate Gaussian distribution with mean 0, variance 1, and covariance $g$. The two-point correlation function of the $Li_2S$ phase is calculated equivalently. $G_g(Y_1,Y_2)$ are obtained via Hermite polynomials as described in Gommes et al.[56]. The angle $\delta$ and the $Li_2S_4$/EL boundary line in Supplementary Fig. 10d–f, defines the morphology of the $Li_2S_2$ phase. For $\delta \to 0$, the $Li_2S_2$ phase will perfectly cover/wet the $Li_2S$ phase in form of a thin film (Supplementary Fig. 10d, g). In contrast, for an $Li_2S_2$/EL boundary parallel to the $Y$-axis ($\delta \to \pi/2$), the $Li_2S_2$ (EL) structure inside the $Li_2S$ cavities is statistically independent from the $Li_2S$ structure (Supplementary Fig. 10f, i). Inserting Eq. 11 into Eqs. 4–5 gives the corresponding scattering intensities (Fig. 4).

## Reporting summary
Further information on research design is available in the Nature Research Reporting Summary linked to this article.

## Data availability
The SANS data generated in this study have been deposited in the ILL database under https://doi.org/10.5291/ILL-DATA.1-04-221. All other data generated in this study are provided within the article and the Supplementary Information file, or are available from the corresponding author C.P. upon request.

## Code availability
The IgorPro (Wavemetrics) code used for SANS data fitting and stochastic modeling are available from the corresponding author C.P. on request.

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

## Acknowledgements

This project has received funding from the European Union's Horizon 2020 research and innovation program under the Marie Skłodowska-Curie grant NanoEvolution, grant agreement No 894042. The authors acknowledge the CERIC-ERIC Consortium for the access to the Austrian SAXS beamline and TU Graz for support through the Lead Project LP-03. Likewise, the use of SOMAPP Lab, a core facility supported by the Austrian Federal Ministry of Education, Science and Research, the Graz University of Technology, the University of Graz, and Anton Paar GmbH is acknowledged. In addition, the authors acknowledge access to the D-22 SANS beamline at the ILL neutron source. Electron microscopy measurements were performed at the Scientific Scenter for Optical and Electron Microscopy (ScopeM) of the Swiss Federal Institute of Technology. C.P. and J.M.M. thank A. Senol for her support with the SANS beamtime preparation. S.D.T., A.V. and R.D. acknowledge the financial support by the Slovenian Research Agency (ARRS) research core

funding P2-0393 and P2-0423. Furthermore, A.V. acknowledge the funding from the Slovenian Research Agency, research project Z2–1863. S.A.F. is indebted to IST Austria for support.

## Author contributions

C.P. carried out operando x-ray and neutron scattering, electrochemical and electron microscopy experiments, the corresponding data analysis and the stochastic modelling. J.M.M. carried out operando neutron scattering, Raman spectroscopy and transmission electron microscopy measurements. S.D.T. and A.V. provided carbon/sulfur composites and polysulfide powders. H.A. supported SAXS measurements, L.P. SANS measurements. C.P., S.A.F. and V.W. conceptualized the work. C.P. wrote the initial version of the manuscript. C.P., J.M.M., S.D.T., A.V., R.D, H.A., L.P., S.A.F. and V. W. contributed to results interpretation and revising the manuscript.

## Competing interests

The authors declare no competing interests.
