## [Peer Review File · Nature Communications]

REVIEWER COMMENTS

Reviewer #1 (Remarks to the Author):

This manuscript provides direct experimental evidence that next to solid Li₂S crystallites, smaller solid short chain Li₂S_x particles are formed upon discharge in Li-S batteries., and suggests that these particles are likely Li₂S₂. However, this study lacks depth and elaboration, which also lacks guidance for the subsequent research on lithium-sulfur batteries. The specific issues that should be fixed are as follows:

1. The main objective of this study is to purge the intermediate products of lithium-sulfur batteries during charging and discharging by operando X-ray/neutron-based techniques. However, the operando SAXS/WAXS experimental data can only show the generation process of Li₂S, even the relationship between intensities of the SAXS and WAXS features and product during the charging and discharging is not clearly articulated.
2. In addition, there have been relevant studies demonstrating the conversion process of the intermediate product Li₂S₂ to Li₂S, such as 10.1021/acsnano.1c00556, 10.1039/c5cp02781k and 10.1002/aenm03638. This studies lack of depth and innovation.
3. The author indicated that the operando X-ray/neutron-based techniques are limited by cell design, the Li₂S stability, the resolution, field of view, or the challenges of 3D imaging. The authors need to point out what is wrong with the previously designed battery structure and what are the advantages of the newly designed battery model.
4. It should be noted that SAXS, WAXS and SANS technologies are suitable for studying those areas in lithium-sulfur batteries, and what are the specific advantages.
5. There are some grammatical errors in the manuscript, such as "Capacity and rate capability are thus influenced by are species solvation, mobility, and applied current density."

Reviewer #2 (Remarks to the Author):

The authors performed operando SANS and SAXS/WAXS measurements on Li-S cells and ex-situ SAXS/WAXS and Raman spectroscopy on samples from the cells. With the results, a mechanism of Li₂S formation via solid-state electrochemical reduction from solid Li₂S₂ is proposed. The proposed final reaction steps of the discharge process would be important information to the research community

since it is still not well described in the literature. However, before pinning down the mechanism, some clarifications about the experimental details and the interpretation of the results are required.

It is demonstrated that the electrochemical properties of the S-cathode/catholyte is sensitive to the electrolyte-to-sulfur (E/S) ratio, sulfur-loading and electrolyte salt and solvent by a large number of literature, e.g. 10.1149/2.0071803jes, 10.1016/j.jpowsour.2014.05.143, 10.1149/1945-7111/abe7a2. Based on the distinct potential profiles, it is reasonable to suspect that the reactions mechanisms vary with the experimental parameters. Therefore, if the authors would like to propose a reaction mechanism, the E/S ratio and sulfur-loading should be clearly stated with the results, preferably where the electrolyte compositions are mentioned.

Following the same concerns above, the authors should also address how the different electrolyte formulations may affect the reactions mechanisms in SANS and SAXS cells. I understand that the options for deuterated electrolyte is rather limited, but the addition of LiNO_3 has been shown to alter the reaction mechanism, 10.1021/acsami.9b07048. Perhaps the authors can demonstrate the similarity/differences of the electrochemical properties of the cells with the two electrolytes, or together with the different cathode setup, by coin cell testing. Indeed, the diverse choices of carbon host in the manuscript also hinders the comparison between scattering results, given the large amount of literature on the effect of carbon matrix on the Li-S cells. The authors probably would like to show the effect of carbon by showing the specific capacity normalised to the mass of carbon. However, this makes it hard to compare the results here to the ones in the literature since the capacity is almost always normalised to the mass of sulfur, which is the active material. This also results in the confusion about how the C-rates are calculated, e.g. in supplementary figure 2.

Regarding the scattering experiments, the scattering results from the cell components, or at least an empty cell, should be examined to make sure 'all the changes observed are entirely cause by the formation of Li_2S_2 or Li_2S ' as the authors state on page 7. Another concern about the experimental groundwork is the calculation of the SLD of the electrolyte. It seems like the electrolyte salts are omitted in the calculations. Considering the large concentrations, would the electrolyte salt not contribute to the SLD?

There are some doubts about the proposed mechanism on page 10 based on the experimental results here. In Figure 2, the authors assign the scattering feature at lower q , q_A , to Li_2S and q_B to Li_2S_2 . However, it is counter-intuitive to see that q_A decreases towards the end of discharge in Figure 2f. Would the average oxidation number of the sulfur species decrease towards the end of discharge? If so, wouldn't the amount of Li_2S be expected to increase towards the end of discharge? The blue line, which shows the diffraction intensity from Li_2S indeed increases towards the end of discharge, contradicting the trend of q_A . The authors argue that q_B is contributed by Li_2S_2 by the SAXS measurement of dried Li_2S solution in THF, but wouldn't such solution contain also Li_2S ? The other argument based on the unwashed and washed electrodes is also not so clear to me. If Li_2S forms in the matrix of Li_2S_2 as

suggested in figure 10, wouldn't it also be washed away or stay with the Li₂S₂ since it is not attached to the carbon surface? Thus, what the authors argue to be washed or not washed away could be either Li₂S or Li₂S₂? Perhaps, if it is possible to do quantitative analysis on the XRD peaks in Figure 3b, it can be shown that how much crystalline Li₂S is left after washing, but it is still unclear if the washed away amorphous scatterer is Li₂S or Li₂S₂. The Raman spectra in Figure 3c is unfortunately unable to tell us if Li₂S₂ is solid or dissolved in the electrolyte while the cell is in operation.

Following the search for Li₂S₂, the authors present the XRD patterns of solid Li and S mixture. It is clear that Li₂S is formed in solid state reactions, but the XRD results are not evidence for the existence of Li₂S₂ precipitates in an operating cell.

Nevertheless, the authors applied a model with three phases, electrolyte, Li₂S₂ and Li₂S to fit the SANS data. Given the similar SLD of Li₂S₂ and Li₂S, it is concerning to see that the trends in the volume fractions of Li₂S and Li₂S₂ are so similar to each other in supplementary Figure 9 f and g. Could it be possible that the results can actually be fitted with just two phases, electrolyte and Li₂S? Indeed, in previous reports, Li₂S has been demonstrated to have different morphologies, [10.1021/acs.chemmater.9b03255](https://doi.org/10.1021/acs.chemmater.9b03255)

All in all, the proposed mechanism is a probable one but there may not be enough proof that Li₂S₂ exists as precipitates in an operating cell. Due to this uncertainty in the mechanism, the title of the manuscript could benefit from some reconsideration.

Some minor suggestions are listed below.

- Supplementary figure 1 does not clearly show the difference between windows and cavities in the cells.
- Supplementary figure 2: the comparisons made in the text are not really feasible with the data presented in heatmaps. It would be better if selected scattering results can be plotted in intensity vs. q .
- Figure 2f: why is peak height (A) shown here instead of integrated intensity?
- As mentioned above, the different experimental parameters of Li-S cells used in different experiments should be stated and possible influence caused by the parameters should be elaborated.
- Figure 3c: the peaks should be labelled more clearly. Is peak around 450 cm⁻¹ Li₂S₄ or Li₂S₂?
- The quality of fitting for SANS data should be presented.

Reviewer #3 (Remarks to the Author):

The article brings light into the mechanism of formation of Li₂S in Li-S batteries by combining operando SAXS/SANS and stochastic modelling. The formation of Li₂S₂ in the pathway of the reduction of sulfur to Li₂S is proposed, and supported by Raman and XRD measurements. I recommend publication of the article as it is.

We thank the Reviewers for their helpful comments to which we respond below point by point. The Reviewers' comments are reproduced in black. In the manuscript and supporting information we highlighted the changes in yellow. The comments helped us greatly to improve the manuscript and we are confident that we have satisfyingly responded to all of them.

Reviewer #1:

This manuscript provides direct experimental evidence that next to solid Li_2S crystallites, smaller solid short chain Li_2S_x particles are formed upon discharge in Li-S batteries., and suggests that these particles are likely Li_2S_2 . However, this study lacks depth and elaboration, which also lacks guidance for the subsequent research on lithium-sulfur batteries. The specific issues that should be fixed are as follows:

1. The main objective of this study is to purge the intermediate products of lithium-sulfur batteries during charging and discharging by operando X-ray/neutron-based techniques. However, the operando SAXS/WAXS experimental data can only show the generation process of Li_2S , even the relationship between intensities of the SAXS and WAXS features and product during the charging and discharging is not clearly articulated.

In the revised manuscript we now explain what SAXS/WAXS and SANS can offer beyond widely used wide angle scattering (XRD). In general, small angle scattering can track the nanoscale structural evolution of any solid (or liquid) with significant materials contrast with respect to its surrounding (e.g., Li_2S with respect to the surrounding electrolyte plus carbon black). Feature sizes accessible are from \sim nm to 100s of nm and phases can also be amorphous. Therefore, SAXS/WAXS and SANS are not only sensitive to Li_2S , but also to other possible polysulfide phases, such as amorphous Li_2S_2 .

Contrary to the integrated intensity of the diffraction peaks in the WAXS regime, the time-dependent intensity changes in the SAXS regime (Fig. 2d, f) are not linearly dependent on the volume fraction of Li_2S . The shape of the SAXS curves depend on the multiphase nanoscale structure and the materials contrasts between all contributing materials phases. The materials contrast is related to scattering length densities (SLDs) as shown in Fig. 4a and is different for X-ray and neutrons. Qualitatively, a SAXS intensity hump around $q \sim 1.5 \text{ nm}^{-1}$ corresponds to a solid phase or particle with a size around $\pi/1.5 \text{ nm}^{-1} \approx 2 \text{ nm}$. For an exact, quantitative analysis, the structure and scattering contrast of the surrounding materials need to be considered as well. A rigorous treatment of the small angle scattering intensity of a three-phase system is given in the Methods section.

Because of the low materials contrasts with X-rays (see Fig. 4a), we discuss the relative SAXS intensity changes only qualitatively in Fig. 2. A quantitative analysis is made in Fig. 4 using SANS and stochastic modeling. Based on the SAXS/WAXS data in Fig. 2 we made the following qualitative statements: i) the Li_2S diffraction peaks in the WAXS signal (Fig. 2b, e) show Li_2S formation during charge and dissolution during discharge. The peak width indicates a Li_2S crystallite size around 7 nm (using the Scherrer

equation). ii) the SAXS intensity maximum in the q_A regime (Fig. 2d) indicates the formation of polycrystalline Li_2S aggregates with a size around 26 nm. This means, the q_A maximum cannot be simply explained by the Li_2S primary crystals. iii) The SAXS intensity maximum in the q_B regime indicates the formation of particles with a size around 2.8 nm. Also, these particles cannot be explained by the primary Li_2S crystals, as observed by the diffraction peaks. Statements i) – iii) lead to the following conclusions: First, there must be more than one solid-discharge product. Second, the Li_2S primary crystals form likely larger aggregates with a size around 26 nm. To elaborate this further, we applied Raman spectroscopy, SANS, stochastic modelling and TEM (newly added in the revised version), as shown in Fig. 3, Fig. 4 of the main manuscript.

2. In addition, there have been relevant studies demonstrating the conversion process of the intermediate product Li_2S_2 to Li_2S , such as 10.1021/acsnano.1c00556, 10.1039/c5cp02781k and 10.1002/aenm03638. This studies lack of depth and innovation.

In the revised manuscript, we now cite these works and specify how our study goes beyond them.

While we agree with the interpretation of those studies, there has hardly been a direct structural or spectroscopic evidence for Li_2S_2 as a solid discharge product in a working *operando* cell. Discussions about the existence of Li_2S_2 are mostly based on the low solubility, DFT simulations or the fact that *operando* XAS found a stoichiometry that corresponds to a mixture of Li_2S and (dissolved) short-chain PSs¹. *Operando* XRD has never shown solid Li_2S_2 after discharge. Ref. 10.1039/c5ta00499c, for example, states in the introduction: *Nevertheless, in situ and operando experiments do not show an obvious signature of Li_2S_2 , which casts doubt on the existence of Li_2S_2 . Li_2S_2 is even deemed as the final discharge product together with Li_2S . Unfortunately, Li_2S_2 has not been isolated from Li–S batteries so far. Thomas and Jones have reported solid Li_2S_2 , but they could not confirm its purity. Subsequently, a less stable Li_2S_4 was also reported, which may exist as a metastable phase. By all appearances, there is a necessity to re-examine the phase diagram of Li–S systems and to unravel the role of Li_2S_2 in Li–S batteries.*

Here, we show that it needed *operando* SAXS/WAXS, *operando* SANS, and stochastic modelling to give a structural evidence for solid Li_2S_2 . It couldn't be found in XRD because the Li_2S_2 particles are small (around 2.8nm) and amorphous. It couldn't be found with *operando* XAS or *operando* Raman spectroscopy because dissolved short-chain polysulfides (always present in the electrolyte) cannot be distinguished from solid ones.

Going beyond previous works, we uniquely quantify the nanoscale structure of the Li_2S_2 phase as a function of time during discharge and charge. This allows drawing detailed conclusions related to the underlying physico-chemical mechanisms. Hence, our work not only quantifies the structural evolution of Li_2S_2 and Li_2S , it also aims to identify the very fundamental mechanism to reversibly convert sulfur (S) into lithium sulfide (Li_2S) and back.

The mechanism presented in Fig. 6, main manuscript shifts paradigms of how to influence the reaction and discharge capacity.

¹R. Dominko et al., *Polysulfides Formation in Different Electrolytes from the Perspective of X-ray Absorption Spectroscopy*, Journal of The Electrochemical Society 165, 1 (2017).

First, previous works^{2,3,4,5} suggested direct electroreduction to form solid Li_2S and hence electron transport through a passivating surface film to limit capacity and rate. Contrary to that, we show mass transport through the tortuous $\text{Li}_2\text{S}/\text{Li}_2\text{S}_2$ network to limit discharge capacity and rate. Factors that influence them are, therefore, species solvation, diffusivities in the electrolyte and the $\text{Li}_2\text{S}_2/\text{Li}_2\text{S}$ solids, and applied current density. Design strategies to improve device performance must therefore change.

Second, the resulting mechanism explains why theoretical sulfur capacities have never been achieved. We cannot fully convert all S into Li_2S ; a certain amount of polysulfides remains as a second solid phase (Li_2S_2). Key to increase practical capacities in the future is to account for the found solid-state conversion.

Another important aspect of the presented work are the novel methods useful for beyond intercalation-type batteries: operando SAXS/WAXS and operando SANS. Below we discuss the unique strengths of these methods.

3. The author indicated that the operando X-ray/neutron-based techniques are limited by cell design, the Li_2S stability, the resolution, field of view, or the challenges of 3D imaging. The authors need to point out what is wrong with the previously designed battery structure and what are the advantages of the newly designed battery model.

Indeed, we have mentioned that (*operando*) electron and X-ray microscopy may be limited by either cell design, the Li_2S stability, the resolution, the field of view, or the challenges of 3D imaging. Importantly each of these methods has its unique strengths and limitations. In many *operando* transmission electron microscopy (TEM) studies for example, a solid electrolyte is used to enable ion transport between cathode and anode material⁶. The S/ Li_2S conversion mechanism in such cell is of course, entirely different from a liquid electrolyte Li-S battery cell. If the spatial resolution with operando X-ray tomography should be high (in the range of a few hundred nanometers), the electrode and separator dimensions need to be particularly small in the order of some tens of micrometers⁷. This makes it difficult to achieve the same electrochemical characteristics as a conventional lab-scale coin cell. SAXS/WAXS and SANS operando cells have the advantage that their cell assembly and electrode separator dimensions are equivalent to those of standard lab-scale coin cells (see Supplementary Fig. 1).

In the revised manuscript, we have extended this discussion and more properly discussed the unique strengths and limitations of each method.

4. It should be noted that SAXS, WAXS and SANS technologies are suitable for studying those areas in lithium-sulfur batteries, and what are the specific advantages.

² S.-Y. Lang et al., *Insight into the Interfacial Process and Mechanism in Lithium–Sulfur Batteries: An In Situ AFM Study*, *Angew. Chem.* 55, 51 (2016).

³ R. Xu, J. Lu, and K. Amine, *Progress in Mechanistic Understanding and Characterization Techniques of Li-S Batteries*, *Advanced Energy Materials* 5, 16 (2015).

⁴ C. Barchasz et al., *New insights into the limiting parameters of the Li/S rechargeable cell*, *Journal of Power Sources* 199 (2012).

⁵ F. Y. Fan, W. C. Carter, and Y.-M. Chiang, *Mechanism and Kinetics of Li_2S Precipitation in Lithium–Sulfur Batteries*, *Adv. Mater.* 27, 35 (2015).

⁶ Z. Yang et al., *Phase Separation of $\text{Li}_2\text{S}_2/\text{S}$ at Nanoscale during Electrochemical Lithiation of the Solid-State Lithium–Sulfur Battery Using In Situ TEM*, *Advanced Energy Materials* 6, 20 (2016).

⁷ P. Pietsch and V. Wood, *X-Ray Tomography for Lithium Ion Battery Research: A Practical Guide*, *Annu. Rev. Mater. Res.* 47, 1 (2017).

To put Me-S batteries into practice, we need to understand and control how the physicochemical mechanisms define the structural evolution of active materials at nanometer length scales. This requires metrologies with a structural sensitivity on these scales; the processes' complexity and transient nature require at least some *operando* capability. Operando SAXS/WAXS/SANS offers integral time-resolved structural information to study complex transient processes. Combined with stochastic modelling, SAXS and SANS allow elucidating otherwise hardly accessible quantitative information about reaction mechanisms, growth processes, and phase transformations in batteries or multiphase energy materials. Due to the complementary materials contrast for SAXS and SANS, even complex multiphase nanoscale structures can be analyzed.

In this study SAXS/WAXS and SANS were particularly useful because of the high spatial resolution, the sensitivity for both crystalline and amorphous solids, and the ability to study the nanoscale structure under practical conditions in an *operando* cell.

We would like to refer to the conclusions, where we discussed the strengths and advantages of SAXS and SANS combined with stochastic modeling. We have also extended the discussion on the relevance of SAXS/WAXS and SANS in the introduction of the revised manuscript.

5. There are some grammatical errors in the manuscript, such as “Capacity and rate capability are thus influenced by are species solvation, mobility, and applied current density.”

Thank you for pointing this out. This has been corrected in the revised manuscript.

Reviewer #2 (Remarks to the Author):

The authors performed operando SANS and SAXS/WAXS measurements on Li-S cells and ex-situ SAXS/WAXS and Raman spectroscopy on samples from the cells. With the results, a mechanism of Li₂S formation via solid-state electrochemical reduction from solid Li₂S₂ is proposed. The proposed final reaction steps of the discharge process would be important information to the research community since it is still not well described in the literature. However, before pinning down the mechanism, some clarifications about the experimental details and the interpretation of the results are required.

We thank reviewer 2 for the very thorough review, which helped us improving where we haven't been clear enough.

It is demonstrated that the electrochemical properties of the S-cathode/catholyte is sensitive to the electrolyte-to-sulfur (E/S) ratio, sulfur-loading and electrolyte salt and solvent by a large number of literature, e.g. 10.1149/2.0071803jes, 10.1016/j.jpowsour.2014.05.143, 10.1149/1945-7111/abe7a2. Based on the distinct potential profiles, it is reasonable to suspect that the reactions mechanisms vary with the experimental parameters. Therefore, if the authors would like to propose a reaction mechanism, the E/S ratio and sulfur-loading should be clearly stated with the results, preferably where the electrolyte compositions are mentioned.

This is indeed an important aspect. The E/S ratios and sulfur loadings are now mentioned where the electrolyte compositions are mentioned. Please note that SAXS/WAXS and SANS measurements have been carried out during potentiostatic charge/discharge of a 0.5 M Li₂S₈ catholyte and a pure carbon electrode. The E/S ratio is thus defined by the Li₂S₈ concentration in the catholyte and corresponds to 7.8 μl/mgs. The highest theoretical sulfur mass loading in the cathode is then defined by the amount of catholyte and corresponds to 19.96 mgs cm⁻².

Following the same concerns above, the authors should also address how the different electrolyte formulations may affect the reactions mechanisms in SANS and SAXS cells.

The proposed reaction mechanism in Fig. 6 is generally valid and can in principle account for the dependency of discharge capacity and deposit morphology on the electrolyte formulation. Li₂S₂ precipitation is likely a solution-mediated process (e.g., Li₂S₄ disproportionation with subsequent precipitation). The electrolyte's donor number or solvation strength (changed for example by the solvent or LiNO₃ supporting salt) would affect solubilities, Li₂S₂ precipitation kinetics and morphology, and in this way the final Li₂S₂/Li₂S aggregate structure. The Li₂S/Li₂S₂ aggregate structure in turn would affect the discharge capacities.

Interestingly, literature^{8,9,10} shows that the morphology on the micrometer scale (as seen by SEM) strongly depends on electrolyte, supporting salt, etc. The Li₂S primary crystallite size, however, (as seen by the XRD peak broadening) is always in the range around 10 nm, independent or only weakly dependent on the electrolyte formulation and other factors. This is a good indicator that the proposed mechanism that leads to the Li₂S/Li₂S₂ composite nanostructure is generally valid for a wide range of systems. The proposed mechanism (Fig. 6) can explain this behavior as the precipitation of Li₂S₂

⁸ Z. Li et al., *Solvent-Mediated Li₂S Electrodeposition: A Critical Manipulator in Lithium–Sulfur Batteries*, *Advanced Energy Materials* 9, 1 (2019).

⁹ B. Yang et al., *Critical Role of Anion Donicity in Li₂S Deposition and Sulfur Utilization in Li-S Batteries*, *ACS Appl. Mater. Interfaces* 11, 29 (2019).

¹⁰ S. Drvarič Talian et al., *Which Process Limits the Operation of a Li–S System?*, *Chemistry of Materials* 31, 21 (2019).

aggregates defines the microscale structure, which is then converted to crystalline Li_2S by a solid-state process. Due to the solid-state process the nanoscale structure is relatively independent of the electrolyte system.

To demonstrate general validity of our results, we also carried out operando SAXS/WAXS during galvanostatic charge/discharge in carbon black/sulfur composite cathodes (Supplementary Fig. 2). There the type of solvent, type of carbon, the E/S ratios and the sulfur mass loading is different.

I understand that the options for deuterated electrolyte is rather limited, but the addition of LiNO_3 has been shown to alter the reaction mechanism, 10.1021/acsami.9b07048. Perhaps the authors can demonstrate the similarity/differences of the electrochemical properties of the cells with the two electrolytes, or together with the different cathode setup, by coin cell testing.

This has possibly been a misunderstanding. We have used 0.4 M LiNO_3 as a supporting salt in both the non-deuterated and the deuterated catholyte. Minor differences in the SAXS and SANS electrochemical data originate from variations during cell assembly, the larger electrodes needed for the SANS operando cells (the neutron beam has a diameter of ca. 10 mm), and the resulting differences in the E/S ratios. Detailed information concerning the SANS electrode dimensions is added to the methods section.

Indeed, the diverse choices of carbon host in the manuscript also hinders the comparison between scattering results, given the large amount of literature on the effect of carbon matrix on the Li-S cells. The authors probably would like to show the effect of carbon by showing the specific capacity normalised to the mass of carbon. However, this makes it hard to compare the results here to the ones in the literature since the capacity is almost always normalised to the mass of sulfur, which is the active material. This also results in the confusion about how the C-rates are calculated, e.g. in supplementary figure 2.

This is a very good point. In fact, we have tested three different carbons: two high-surface area carbon blacks, Ketjen black (Fig. 2, Fig. 4), ENSACO (Supplementary Fig. 2) and large glassy carbon beads (Supplementary Fig. 10). For all carbons the main features in the SAXS/SANS signal are present after discharge: (i), The high- q (q_B) intensity shoulder corresponding to the Li_2S_2 particles, and (ii), the low- q (q_A) intensity shoulder corresponding to the Li_2S aggregates.

In the table below we summarize the investigated systems: type of carbon cathodes, the type of electrolytes, the E/S ratios, and the S mass loadings. For all systems, we found the SAXS/WAXS/SANS features indicative for Li_2S nanocrystals/aggregates, and smaller amorphous Li_2S_2 nanoparticles.

	Fig. 2	Fig. 4	Suppl. Fig. 10	Suppl. Fig. 2a	Suppl. Fig. 2b	Suppl. Fig. 2c
Discharge /charge	potentiostatic 2.0/2.45 V	potentiostatic 2.0/2.45 V	potentiostatic 2.0/2.45 V	galvanostatic C/3	galvanostatic C/10	galvanostatic C/30
Method	SAXS/WAXS	SANS	SAXS/WAXS	SAXS/WAXS	SAXS/WAXS	SAXS/WAXS
Cathode	KB	KB	GC	ENSACO/S 1:2	ENSACO/S 1:2	ENSACO/S 1:2
Catholyte/ Electrolyte	0.5 M Li_2S_8 + 1 M LiTFSI + 0.4 M LiNO_3 in 2G	0.5 M Li_2S_8 + 1 M LiTFSI + 0.4 M LiNO_3 in 2G deuterated	0.5 M Li_2S_8 + 1 M LiTFSI + 0.4 M LiNO_3 in 2G	1 M LiTFSI + 0.1 M LiNO_3 in 2G	1 M LiTFSI + 0.1 M LiNO_3 in 2G	1 M LiTFSI + 0.1 M LiNO_3 in 2G

E/S ratio ($\mu\text{L mg s}^{-1}$)	7.81	7.81	7.81	156.25	143.44	162.04
S mass (mg)	7.68	25.6	7.68	1.12	1.22	1.08
S mass loading (g cm^{-2})	19.96	19.29	19.96	2.23	2.43	2.15

Considering all these parameters, we believe that our findings are valid for a relatively wide range of systems. We have discussed this in the revised version manuscript and added the table to the SI.

Regarding the scattering experiments, the scattering results from the cell components, or at least an empty cell, should be examined to make sure 'all the changes observed are entirely cause by the formation of Li_2S_2 or Li_2S ' as the authors state on page 7.

We have added the empty cell measurements of SAXS/WAXS cells and background SANS measurements (without cathode) to Supplementary Fig. 4. SAXS/WAXS intensities of separator plus Li metal anode after discharge (Supplementary Fig. 3) show that the measured intensity changes are indeed caused by changes in the cathode only. The anode, separator, and empty cell contribute only with a constant background.

Another concern about the experimental groundwork is the calculation of the SLD of the electrolyte. It seems like the electrolyte salts are omitted in the calculations. Considering the large concentrations, would the electrolyte salt not contribute to the SLD?

The SAXS SLD for the pure 2G solvent corresponds to $0.874 \times 10^{11} \text{ cm}^{-2}$, for 1 M LiTFSI in 2G corresponds to $0.914 \times 10^{11} \text{ cm}^{-2}$, 0.4 M LiNO_3 in 2G corresponds to $0.891 \times 10^{11} \text{ cm}^{-2}$, 0.5 M Li_2S_8 in 2G corresponds to $0.924 \times 10^{11} \text{ cm}^{-2}$. The catholyte (0.5 M Li_2S_8 in 1 M LiTFSI + 0.4 M LiNO_3 in 2G) SLD corresponds to $0.98 \times 10^{11} \text{ cm}^{-2}$, i.e. not substantially higher than the pure 2G solvent. Changes in salt and polysulfide concentration during charging and discharging would lead to SLD changes in the order of a few %. Since the SLD contrasts between carbon, electrolyte, and Li_2S are small for x-rays, these changes can be visible in the SAXS signal. We have discussed this in response to the next question.

The SANS SLDs are essential for the quantitative analysis in Fig. 4. Here, the contrast between the $\text{Li}_2\text{S}/\text{Li}_2\text{S}_2$ SLD and the surrounding carbon/electrolyte is large. Hence, changes in the electrolyte SLD (in the order of a few %) due to ion and polysulfide concentration changes have a negligible effect on the SANS intensities during $\text{Li}_2\text{S}/\text{Li}_2\text{S}_2$ formation. For the SANS model fit, we assumed that $\text{SLD}_{\text{EL}} \approx \text{SLD}_{\text{C}} = 6.67 \times 10^{10} \text{ cm}^{-2}$.

As shown in Fig. 4b, the SANS intensity increases about two orders of magnitude during discharge. The formation of the $\text{Li}_2\text{S}/\text{Li}_2\text{S}_2$ structures dominates the SANS intensity changes.

There are some doubts about the proposed mechanism on page 10 based on the experimental results here. In Figure 2, the authors assign the scattering feature at lower q , q_A , to Li_2S and q_B to Li_2S_2 . However, it is counter-intuitive to see that q_A decreases towards the end of discharge in Figure 2f. Would the average oxidation number of the sulfur species decrease towards the end of discharge? If so, wouldn't the amount of Li_2S be expected to increase towards the end of discharge?

In general, the shape of the SAXS curves does not simply depend on the amount of solid Li_2S_2 particles and Li_2S aggregates. The exact SAXS intensity changes are a complex function of all contributing structures and their cross-correlations (see Equ. 5-6 for a three-phase system). Next to the phases' individual scattering contribution, also the SLD differences (contrasts) can change during charging and discharging. The decrease of the SAXS intensity in the q_A regime could be explained by a slight increase of the dissolved PS concentration in the electrolyte. We have explained this behavior in the revised version of the manuscript.

Important for the qualitative interpretation of the SAXS data in Fig. 2 are the presence of two intensity maxima, occurring during discharge and disappearing during charge. To interpret the data quantitatively and to see the impact of $\text{Li}_2\text{S}/\text{Li}_2\text{S}_2$ formation alone, we performed operando SANS with deuterated electrolyte. As explained in the response above, SANS minimizes the carbon scattering contribution because $\text{SLD}_{\text{EL}} \approx \text{SLD}_{\text{C}}$ and it makes ion and PS concentration changes negligible because $\Delta(\text{SLD}_{\text{EL}} - \text{SLD}_{\text{Li}_2\text{S}})$ and $\Delta(\text{SLD}_{\text{EL}} - \text{SLD}_{\text{Li}_2\text{S}_2})$ are large.

The blue line, which shows the diffraction intensity from Li_2S indeed increases towards the end of discharge, contradicting the trend of q_A .

Another important aspect is that the q_A SAXS intensity does not scale linearly with the Li_2S amount. While the diffraction peak in the WAXS signal records the formation of the Li_2S primary crystals the q_A intensity maximum relates to the morphology of the Li_2S aggregates (consisting of several Li_2S primary crystals as visualized in Fig. 4d).

The authors argue that q_B is contributed by Li_2S_2 by the SAXS measurement of dried Li_2S solution in THF, but wouldn't such solution contain also Li_2S ?

In Fig. 3a, we referred to a dried Li_2S_2 solution. Before drying the solution was diluted until all Li_2S could be dissolved. Indeed, a stoichiometric Li_2S_2 solution can, in principle, contain several polysulfides. If all polysulfides are dissolved, the largest fraction should still be Li_2S_2 . If Li_2S were present in the Li_2S_2 solution, given its low solubility, it would be present as a solid (nanocrystal). According to the XRD, there is no crystalline Li_2S (blue curve in Fig. 3b). We concluded that the SAXS intensity shoulder in Fig. 3a should stem from Li_2S_2 nanoparticles.

The other argument based on the unwashed and washed electrodes is also not so clear to me. If Li_2S forms in the matrix of Li_2S_2 as suggested in figure 10, wouldn't it also be washed away or stay with the Li_2S_2 since it is not attached to the carbon surface? Thus, what the authors argue to be washed or not washed away could be either Li_2S or Li_2S_2 ?

According to our mechanism, Li_2S is formed via solid-state conversion of Li_2S_2 to Li_2S . If this conversion is a direct electroreduction, a significant amount of Li_2S should be in direct contact with and attached to the carbon surface. We think that the Li_2S aggregates are widely interconnected; washing away a large fraction of Li_2S_2 , without washing away the Li_2S should be possible. The washing step involves gently rinsing the electrode with about 1 mL of 2G. Washing away the interconnected Li_2S aggregates with such a step seems unlikely, considering the tortuous and about 100 nm large pores of the KB electrode.

Perhaps, if it is possible to do quantitative analysis on the XRD peaks in Figure 3b, it can be shown that how much crystalline Li₂S is left after washing, but it is still unclear if the washed away amorphous scatterer is Li₂S or Li₂S₂. The Raman spectra in Figure 3c is unfortunately unable to tell us if Li₂S₂ is solid or dissolved in the electrolyte while the cell is in operation.

If Li₂S could be washed away, we should be able to wash away both crystalline and hypothetical amorphous Li₂S. Within the accuracy of the measurement the diffraction peak intensity of crystalline Li₂S in washed and unwashed electrode are the same (see overlaid light and dark grey XRD curve in the Figure below). The high-q SAXS intensity shoulder of Li₂S₂ on the other hand disappears after washing. This suggests that the high-q SAXS intensity shoulder is caused by a polysulfide other than Li₂S.

Following the search for Li₂S₂, the authors present the XRD patterns of solid Li and S mixture. It is clear that Li₂S is formed in solid state reactions, but the XRD results are not evidence for the existence of Li₂S₂ precipitates in an operating cell.

That is true. With this experiment we wanted to clarify whether the kinetics for a solid-state conversion and mass transport could in principle be fast enough to convert larger Li₂S₂ aggregates. Indeed, the kinetics is faster than the poor bulk ionic and electronic conductivities would suggest^{11,12}. Likely due to the accelerated transport on interfaces. The evidences for solid Li₂S₂ come from the SAXS/SANS data.

Nevertheless, the authors applied a model with three phases, electrolyte, Li₂S₂ and Li₂S to fit the SANS data. Given the similar SLD of Li₂S₂ and Li₂S, it is concerning to see that the trends in the volume fractions of Li₂S and Li₂S₂ are so similar to each other in supplementary Figure 9 f and g. Could it be possible that the results can actually be fitted with just two phases, electrolyte and Li₂S? Indeed, in previous reports, Li₂S has been demonstrated to have different morphologies, 10.1021/acs.chemmater.9b03255

We thank reviewer 2 for pointing this out. The morphologies in the mentioned work refer primarily to the morphology on micrometer length scales. Here we discuss mainly the structure on nanometer length scales (Fig. 4).

We carried out transmission electron microscopy (TEM) on a C/Au TEM grid after polysulfide electrodeposition in an electrochemical cell. The TEM measurements clearly show the existence of two

¹¹ S. Lorget, R. E. Usiskin, and J. Maier, *Transport and Charge Carrier Chemistry in Lithium Sulfide*, *Advanced Functional Materials* 29, 6 (2019).

¹² R. Usiskin and J. Maier, *Guidelines for Optimizing the Architecture of Battery Insertion Electrodes with Ohmic Surface, Coating, or Electrolyte Resistances*, *Journal of The Electrochemical Society* 167, 8 (2020).

different phases: i) Li_2S nanocrystals and ii) a second amorphous nanoparticulate structure (which we identified as Li_2S_2). The fact that we find such a structure even after electrodeposition on a TEM grid indicates that the existence of the second phase is not restricted to the specific system used for the SAXS and SANS measurements. We have added the TEM data to the revised manuscript.

The second amorphous phase cannot be Li_2S and is likely Li_2S_2 , due to the following reasons:

- i) Li_2S is insoluble in ether-based solvents. While the amorphous phase could be washed away (Fig. 3a), the nanocrystalline Li_2S remained after washing (Fig. 3b). The fact that Li_2S_2 can be washed away also fits the essential experiment in the mentioned paper 10.1021/acs.chemmater.9b03255: After the end of galvanostatic discharge and a washing step in between, even more material could be deposited on the discharged electrode using a fresh catholyte. The washing step enables new partial access to the carbon electrolyte interface, which allows for further discharge.
- ii) If the amorphous nanoparticulate phase would be Li_2S , it should show a similar time-dependency to the nanocrystalline Li_2S during electrochemical charge/discharge. The amorphous phase (Li_2S_2) grows during the initial stages of charge, while the Li_2S aggregates dissolve relatively quickly during charge (Li_2S). Note that this behavior is not only a result of the SANS fit. The q_B intensity maximum in the SAXS signal also points to this initial Li_2S_2 growth during charge. From an electrochemical perspective, this makes only sense if the amorphous phase is formed via a chemical reaction (Li_2S_4 DISP, as suggested in Fig. 6) and if the S oxidation state in the amorphous phase is more positive than in Li_2S (S^{2-}).
- iii) If the same compound (Li_2S) occurs in two different forms ($\sim 7\text{nm}$ nanocrystals and $\sim 2.8\text{ nm}$ amorphous particles), there must be two different formation mechanisms. Indeed, we considered whether nanocrystalline Li_2S would be formed via a solution-mediated mechanism, whereas the amorphous hypothetical Li_2S phase could be formed via a surface mechanism (direct reduction at the carbon-electrolyte interface). However, in that case, the vast majority of the amorphous phase should be in relatively close proximity to the carbon surface. This is not the case, as shown in Fig. 4 and in the TEM image on the top.
- iv) Both SANS (Fig. 4) and SAXS data (Supplementary Fig. 10) were fitted successfully with the $\text{Li}_2\text{S}/\text{Li}_2\text{S}_2$ Plurigaussian Random field model, despite entirely different materials contrasts. For the SAXS fit we had to use a discharged electrode with large glassy carbon beads. Due to the large size of the GC beads their scattering contribution could be simply subtracted. What remains is the contribution of the $\text{Li}_2\text{S}/\text{Li}_2\text{S}_2$ structure.
- v) The ex-situ Raman measurements in Fig. 3c and Supplementary Fig. 7 and the lower solubility of Li_2S_2 compared to longer-chain polysulfides further indicate that the second phase is Li_2S_2 .

All in all, the proposed mechanism is a probable one but there may not be enough proof that Li_2S_2 exists as precipitates in an operating cell. Due to this uncertainty in the mechanism, the title of the manuscript could benefit from some reconsideration.

We agree with reviewer 2, that there is still a lot to be investigated in future works and the title could be more specific. To acknowledge for this, we changed the title to “Mechanism of Li_2S formation and dissolution in Lithium-Sulfur batteries” to “On the nanoscale structural evolution of solid discharge products in lithium-sulfur batteries using neutron, x-ray, and electron techniques”.

Some minor suggestions are listed below.

- Supplementary figure 1 does not clearly show the difference between windows and cavities in the cells.

That's right. We have revised the sketches of SAXS and SANS cells accordingly. Please note that the Aluminum windows in the SANS cells are nothing else but cavities in the upper and bottom cell parts.

- Supplementary figure 2: the comparisons made in the text are not really feasible with the data presented in heatmaps. It would be better if selected scattering results can be plotted in intensity vs. q .

Unfortunately, on a log-log plot (just like in Fig. 2a), the essential features are not visible. The SAXS data in Supplementary Fig. 2 were recorded on a laboratory SAXS facility, making the signal-to-noise ratio much worse than the data in Fig. 2. However, in the relative SAXS intensity the essential features are visible: (i) a maximum in the relative SAXS intensity in the q_B regime after discharge, and (ii) a maximum in the relative SAXS intensity in the q_A regime after discharge, which depends strongly on the C-rate. At C/30 this maximum is missing, likely because the polycrystalline Li_2S aggregates are much larger than the found 26 nm for potentiostatic discharge. Note that the intensity maximum before discharge (and after charge for C/10) is caused by the solid sulfur.

- Figure 2f: why is peak height (A) shown here instead of integrated intensity?

There is no specific reason for that. As the peak width remains relatively constant, the integrated intensity change is nearly identical to the peak height change. However, as we discuss the Li_2S formation based on the diffraction peak, we have replaced the peak height with the peak integrated intensity in the revised manuscript.

- As mentioned above, the different experimental parameters of Li-S cells used in different experiments should be stated and possible influence caused by the parameters should be elaborated.

We have stated the important parameters in Supplementary Table 3 and added a brief discussion, as mentioned above.

- Figure 3c: the peaks should be labelled more clearly. Is peak around 450 cm^{-1} Li_2S_4 or Li_2S_2 ?

We have changed the labeling. As mentioned in the text it is likely Li_2S_2 (peak at 440 cm^{-1}). However, due to the noisy data and the relatively broad peak, we cannot exclude the contribution of Li_2S_4 (peak

at 450 cm^{-1}). However, if the majority would be Li_2S_4 , we would expect a significant Li_2S_4 peak also at 330 cm^{-1} (see Li_2S_4 in 2G, grey curve in Fig. 3c).

- The quality of fitting for SANS data should be presented.

We now show all SANS data fits in Supplementary Fig. 8.

Reviewer #3 (Remarks to the Author):

The article brings light into the mechanism of formation of Li_2S in Li-S batteries by combining operando SAXS/SANS and stochastic modelling. The formation of Li_2S_2 in the pathway of the reduction of sulfur to Li_2S is proposed, and supported by Raman and XRD measurements. I recommend publication of the article as it is.

We thank Reviewer 3 for the positive feedback.

REVIEWER COMMENTS

Reviewer #1 (Remarks to the Author):

In the revised manuscript titled "Mechanism of Li₂S formation and dissolution in Lithium-Sulfur batteries", the authors have addressed all my questions and concerns. The quality of the manuscript has been improved a lot. I don't have further questions.

Reviewer #2 (Remarks to the Author):

The authors have made substantial effort to revise the manuscript according to my previous comments. All in all, I think my comments are addressed, except for the concern about excluding the contribution of Li in the calculation of the neutron SLD of the deuterated electrolyte. I am not sure if I share the same estimation that the authors concluded in the rebuttal letter:

Hence, changes in the electrolyte SLD

(in the order of a few %) due to ion and polysulfide concentration changes have a negligible effect on the SANS intensities during Li₂S/Li₂S₂ formation.

After all, 1.9 M of Li, which has a negative neutron SLD, is added to the electrolyte, which can be expected to lower the SLD of the electrolyte. In addition, the authors set the SLD of the electrolyte and carbon to be the same.

For the SANS model fit, we assumed that $SL_{DEL} \approx$

$SL_{DC} = 6.67 \times 10^{10} \text{ cm}^{-2}$.

Thus, if the SLD of the electrolyte changes during cycling, could there be extra scattering contributions from the electrolyte-filled highly-porous carbon?

Nevertheless, without precise density measurements, I cannot verify if the above concern is actually valid. Therefore, if the authors can confirm their assumptions, I think this manuscript can be published as it is.

We thank the Reviewers for positive feedback. The response to the remaining issue raised by reviewer #2 is given below. The Reviewers' comments are reproduced in black. In the manuscript and supplementary information we highlighted the changes in yellow.

Reviewer #2:

The authors have made substantial effort to revise the manuscript according to my previous comments. All in all, I think my comments are addressed, except for the concern about excluding the contribution of Li in the calculation of the neutron SLD of the deuterated electrolyte. I am not sure if I share the same estimation that the authors concluded in the rebuttal letter:

Hence, changes in the electrolyte SLD (in the order of a few %) due to ion and polysulfide concentration changes have a negligible effect on the SANS intensities during $\text{Li}_2\text{S}/\text{Li}_2\text{S}_2$ formation. After all, 1.9 M of Li, which has a negative neutron SLD, is added to the electrolyte, which can be expected to lower the SLD of the electrolyte. In addition, the authors set the SLD of the electrolyte and carbon to be the same. For the SANS model fit, we assumed that $\text{SLD}_{\text{EL}} \approx \text{SLD}_{\text{C}} = 6.67 \times 10^{10} \text{ cm}^{-2}$.

Thus, if the SLD of the electrolyte changes during cycling, could there be extra scattering contributions from the electrolyte-filled highly-porous carbon? Nevertheless, without precise density measurements, I cannot verify if the above concern is actually valid. Therefore, if the authors can confirm their assumptions, I think this manuscript can be published as it is.

We have now calculated the catholyte's scattering length density (SLD) for different polysulfide concentrations (Fig. 1a) ¹. Based on these values and SANS model calculations (Fig. 1b), we confirm the validity of our assumptions and explain why any extra scattering contributions are negligibly small.

The SLD of the deuterated catholyte is $5.3\text{e}+10 \text{ cm}^{-2}$, which is indeed a little smaller than the SLD of the carbon (Fig. 1a). To estimate the intensity contribution let's assume a two-phase system, like carbon black in deuterated electrolyte. In such system, the SANS intensity scales with the square of the SLD difference (ΔSLD^2) between the two phases. We have calculated the ΔSLD^2 for carbon and catholyte ($1.86\text{e}+20 \text{ cm}^{-4}$), as well as Li_2S and catholyte ($3.03\text{e}+21 \text{ cm}^{-4}$). Hence, the SANS scattering contribution of carbon is only about 6 % of the scattering contribution of solid Li_2S . Since the small remaining carbon scattering background is subtracted from all measured SANS intensities prior to data fitting in Fig. 4b, c (see methods, background subtraction), any remaining error could only stem from second-order effects like cross-correlations between the carbon and $\text{Li}_2\text{S}/\text{Li}_2\text{S}_2$ structure. Such second-order error would be even smaller than the mentioned 6 % of the SANS intensity.

¹ V. F. Sears, *Neutron scattering lengths and cross sections*, Neutron News 3, 3 (1992).

The blue data points in Fig. 1 show the catholyte SLD for different polysulfide (Li_2S_8) concentrations. Starting from the 0.5 M Li_2S_8 concentration with an SLD of $5.3 \times 10^{10} \text{ cm}^{-2}$, the SLD would decrease by a few %, when the Li_2S_8 concentration is increased. Even if the polysulfide concentration would increase by 200 % (from 0.5 M to 1.5 M), the effect on the SANS intensity is absolutely minor. Modeled SANS intensities with the same $\text{Li}_2\text{S}/\text{Li}_2\text{S}_2$ structure but different polysulfide concentrations in the catholyte confirm this (Fig. 1b).

Overall, effects related to possible polysulfide concentration changes or related to our SLD assumptions are negligibly small compared to the huge experimental SANS intensity changes. Note that the SANS intensity increases by two orders of magnitude upon solid $\text{Li}_2\text{S}/\text{Li}_2\text{S}_2$ formation (see Fig. 4b).

Figure 1: a, Scattering length densities (SLDs) vs. Li_2S_8 molarity in the deuterated catholyte. The dashed lines indicate the SLD of other components in the system. **b**, Modelled SANS intensities for the $\text{Li}_2\text{S}/\text{Li}_2\text{S}_2$ structure at the end of discharge. Structural parameters are taken from Table S1, only the catholyte SLDs are different. The black curve uses the SLD for 0.5 M Li_2S_8 , the blue line the SLD for 1.5 M Li_2S_8 . The SANS intensities are only shifted by a small constant factor.

We have added this discussion plus figures to the Supplementary Information.

REVIEWERS' COMMENTS

Reviewer #2 (Remarks to the Author):

I appreciate the effort that the authors put into Figure 12 in the SI and the discussions about it.

I would just like to clarify my previous concern that if the change in the SLD of the catholyte is similar to that of the difference between the SLD of Li_2S and Li_2S_2 as shown in Figure 12, would it be appropriate to fit the SANS data with a three-phase model as laid out in eq. 5 and 6?

However, since the authors supplemented their argument with other experimental observations and also agreed that more data are required for pinning down the exact mechanism, I think this manuscript can be published as it is.

The response to reviewer #2 is given below. The Reviewer's comments are reproduced in black.

Reviewer #2:

I appreciate the effort that the authors put into Figure 12 in the SI and the discussions about it. I would just like to clarify my previous concern that if the change in the SLD of the catholyte is similar to that of the difference between the SLD of Li_2S and Li_2S_2 as shown in Figure 12, would it be appropriate to fit the SANS data with a three-phase model as laid out in eq. 5 and 6?

However, since the authors supplemented their argument with other experimental observations and also agreed that more data are required for pinning down the exact mechanism, I think this manuscript can be published as it is.

We thank Reviewer #2 for the positive feedback. The reason why we need a three-phase model, is not necessarily because Li_2S and Li_2S_2 have a different SLD. We need a three-phase model because the two solid phases have a different structure. The Li_2S_2 particles are significantly smaller than the Li_2S aggregates, and they are not entirely randomly distributed with respect to the Li_2S aggregates. Mathematically, such complex structure could not be treated with a two-phase model, even if the SLDs of Li_2S and Li_2S_2 would be identical.